# Mutational resilience of antiviral restriction favors primate TRIM5α in host-virus evolutionary arms races

**Jeannette L Tenthorey[1]\*, Candice Young[1], Afeez Sodeinde[1], Michael Emerman[1,2]†, Harmit S Malik[1,3]†**

[1]Division of Basic Sciences, Fred Hutchinson Cancer Research Center, Seattle, United States; [2]Division of Human Biology, Fred Hutchinson Cancer Research Center, Seattle, United States; [3]Howard Hughes Medical Institute, Fred Hutchinson Cancer Research Center, Seattle, United States

**Abstract** Host antiviral proteins engage in evolutionary arms races with viruses, in which both sides rapidly evolve at interaction interfaces to gain or evade immune defense. For example, primate TRIM5α uses its rapidly evolving 'v1' loop to bind retroviral capsids, and single mutations in this loop can dramatically improve retroviral restriction. However, it is unknown whether such gains of viral restriction are rare, or if they incur loss of pre-existing function against other viruses. Using deep mutational scanning, we comprehensively measured how single mutations in the TRIM5α v1 loop affect restriction of divergent retroviruses. Unexpectedly, we found that the majority of mutations increase weak antiviral function. Moreover, most random mutations do not disrupt potent viral restriction, even when it is newly acquired via a single adaptive substitution. Our results indicate that TRIM5α's adaptive landscape is remarkably broad and mutationally resilient, maximizing its chances of success in evolutionary arms races with retroviruses.

**\*For correspondence:**
jtenthor@fredhutch.org

†These authors contributed equally to this work

**Competing interests:** The authors declare that no competing interests exist.

## Introduction

Mammalian genomes combat the persistent threat of viruses by encoding a battery of cell-intrinsic antiviral proteins, termed restriction factors, that recognize and inhibit viral replication within host cells. The potency of restriction factors places selective pressure on viruses to evade recognition in order to complete replication (*Duggal and Emerman, 2012*). In turn, viral escape spurs adaptation of restriction factors, by selecting for variants that re-establish viral recognition and thereby restriction (*McCarthy et al., 2015*). Mutual antagonism between viruses and their hosts thus drives cycles of recurrent adaptation, in prey-predator-like genetic arms races (*Van Valen, 1973*). These arms races result in the rapid evolution of restriction factors, which accumulate amino acid mutations at their virus-binding interfaces at a higher than expected rate (*Daugherty and Malik, 2012*).

Numerous restriction factors, including TRIM5α (*Sawyer et al., 2005*), APOBEC3G (*Sawyer et al., 2004*), and MxA (*Mitchell et al., 2012*), evolve rapidly as a result of arms races with target viruses. The resulting divergence between restriction factor orthologs can result in cross-species barriers to viral infection (*Compton and Emerman, 2013*; *Kirmaier et al., 2010*). Such barriers led to the initial identification of TRIM5α, during a screen for proteins that prevented HIV-1 (human immunodeficiency virus) from efficiently replicating in rhesus macaque cells (*Stremlau et al., 2004*). Rhesus TRIM5α could potently restrict HIV-1, whereas the virus almost completely escapes TRIM5α-mediated inhibition in its human host. Subsequent studies revealed that restriction of SIVs (simian immunodeficiency viruses) also varies across TRIM5α orthologs and that SIVs likely drove the rapid evolution of TRIM5α in Old World monkeys (*McCarthy et al., 2015*; *Wu et al., 2013*).

**eLife digest** The evolutionary battle between viruses and the immune system is essentially a high-stakes arms race. The immune system makes antiviral proteins, called restriction factors, which can stop the virus from replicating. In response, viruses evolve to evade the effects of restriction factors. To counter this, restriction factors evolve too, and the cycle continues. The challenge for the immune system is that mammals do not evolve as fast as viruses. How then, in the face of this disadvantage, can the immune system hope to keep pace with viral evolution?

One human antiviral protein that seems to have struggled to keep up is TRIM5α. In rhesus macaques, it is very effective at stopping the replication of HIV-1 and related viruses. But in humans, it is not effective at all. But why? Protein evolution happens due to small genetic mutations, but not every mutation makes a protein better. If a protein is resilient, it can tolerate lots of neutral or negative mutations without breaking, until it mutates in a way that makes it better. But, if a protein is fragile, even small changes can render it completely unable to do its job. It is possible that restriction factors, like TRIM5α, are evolutionarily 'fragile', and therefore easy to break. But it is difficult to test whether this is the case, because existing mutations have already passed the test of natural selection. This means that either the mutation is somehow useful for the protein, or that it is not harmful enough to be removed.

Tenthorey et al. devised a way to introduce all possible changes to the part of TRIM5α that binds to viruses. This revealed that TRIM5α is not fragile; most random mutations increased, rather than decreased, the protein's ability to prevent viral infection. In fact, it appears it would only take a single mutation to make TRIM5α better at blocking HIV-1 in humans, and there are many possible single mutations that would work. Thus, it would appear that human TRIM5α can easily gain the ability to block HIV-1. The next step was to find out whether these gains in antiviral activity are just as easily lost. To do this, Tenthorey et al. performed the same tests on TRIM5α from rhesus macaques and an HIV-blocking mutant version of human TRIM5α. This showed that the majority of random mutations do not break TRIM5α's virus-blocking ability. Thus, TRIM5α can readily gain antiviral activity and, once gained, does not lose it easily during subsequent mutation.

Antiviral proteins like TRIM5α engage in uneven evolutionary battles with fast-evolving viruses. But, although they are resilient and able to evolve, they are not always able to find the right mutations on their own. Experiments like these suggest that it might be possible to give them a helping hand. Identifying mutations that help human TRIM5α to strongly block HIV-1 could pave the way for future gene therapy. This step would demand significant advances in gene therapy efficacy and safety, but it could offer a new way to block virus infection in the future.

TRIM5α disrupts retroviral replication early in infection by binding to the capsid core of retroviruses entering the cell (*Li et al., 2016*; *Maillard et al., 2007*; *Owens et al., 2003*). Its binding causes the premature uncoating of the viral core (*Stremlau et al., 2006*), preventing delivery of the viral genome to the nucleus for integration. TRIM5α binds to the capsid via the unstructured v1 loop within its B30.2 domain (*Biris et al., 2012*; *Sebastian and Luban, 2005*). Experiments swapping the v1 loop between TRIM5α orthologs indicated that it is critical for recognition of capsid from many retroviruses (*Ohkura et al., 2006*; *Perron et al., 2006*; *Sawyer et al., 2005*). In Old World monkeys and hominoids, rapid evolution of TRIM5α is concentrated within this v1 loop (*Sawyer et al., 2005*; *Figure 1A*). Single amino acid mutations at these rapidly evolving sites can cause dramatic gains of restriction against HIV-1 and other retroviruses (*Li et al., 2006*; *Maillard et al., 2007*; *Yap et al., 2005*). However, it remains unclear whether such adaptive mutations are rare among all single mutational steps that might be randomly sampled during TRIM5α's natural evolution.

The functional consequence of all single mutations from a given protein sequence can be visualized as an evolutionary landscape (*Smith, 1970*), in which mutations are either beneficial (fitness peak), detrimental (fitness valley), or neutral. The topology of this evolutionary landscape, in terms of numbers of peaks and valleys and their connections, represents the adaptive potential of restriction factors in their evolutionary arms race with viruses. Previous studies that have empirically mapped evolutionary landscapes of conserved enzymes and transcription factors revealed that ligand-binding residues are highly intolerant to substitutions (*Fowler et al., 2010*; *Guo et al., 2004*;

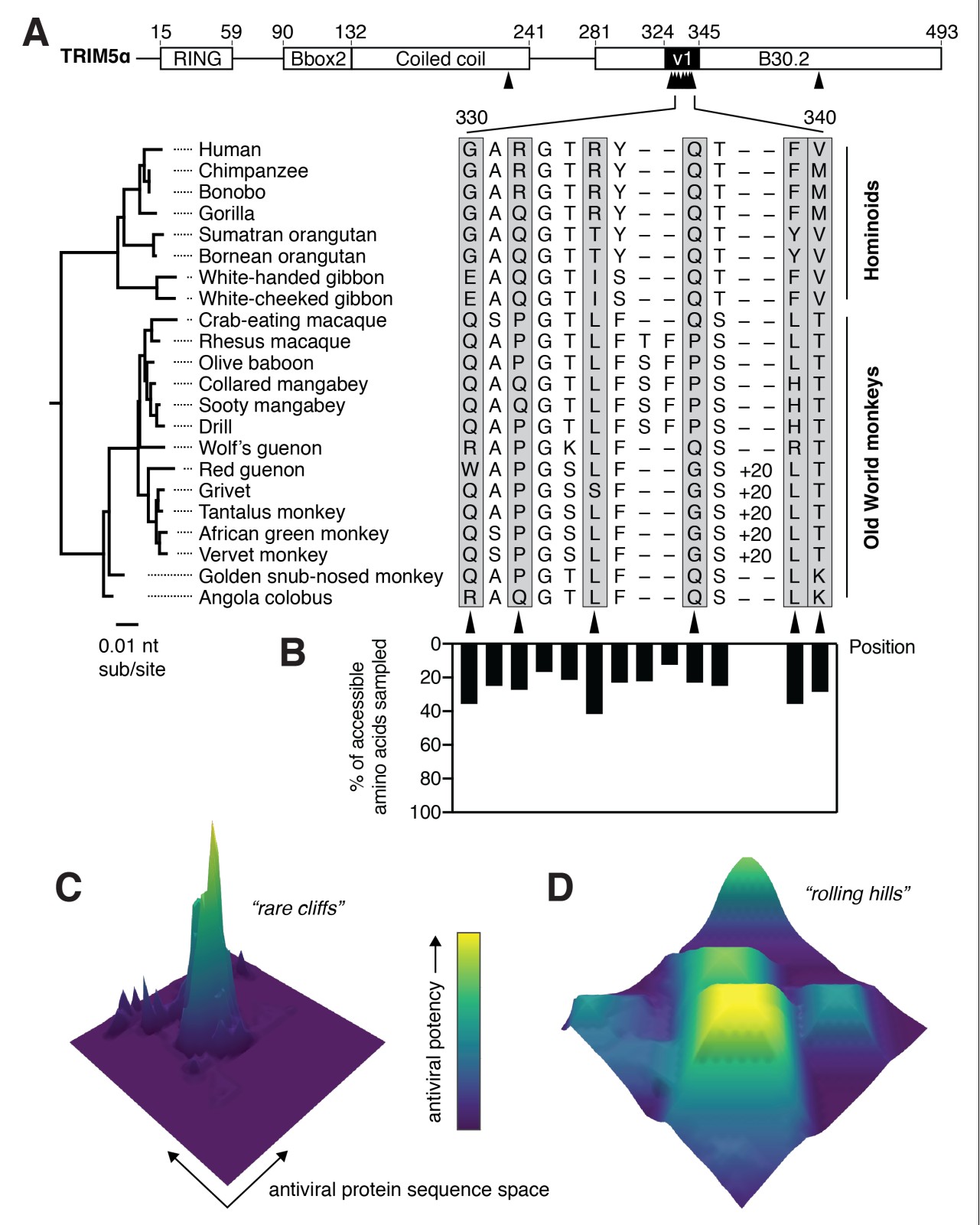

**Figure 1.** TRIM5α has sampled limited amino acid diversity, even at rapidly evolving positions. (**A**) Alignment of TRIM5α from simian primates. A 20-amino acid duplication in the v1 loop of the African green monkey clade is abbreviated as '+20'. Amino acid numbering follows human TRIM5α. Rapidly evolving residues are indicated with black arrows and gray boxes. (**B**) Evolutionarily accessible amino acids were defined as within 1 nucleotide of any codon in this alignment, and the fraction of accessible variants sampled among aligned sequences was determined for each position. (**C-D**)

*Figure 1 continued on next page*

*Figure 1 continued*

Theoretical possibilities for antiviral protein evolutionary landscapes, with antiviral potency represented in z and color axes as it varies with single point mutations. Fitness landscapes might be highly constrained (C) or permissive (D).

The online version of this article includes the following source data for figure 1:

**Source data 1.** Codons that are evolutionarily accessible to primate TRIM5α.

*McLaughlin et al., 2012*; *Suckow et al., 1996*). Moreover, mutations that allowed proteins to gain novel ligand specificity, even for closely related ligands, were rare among all possible substitutions (*McLaughlin et al., 2012*; *Starr et al., 2017*; *Stiffler et al., 2015*). In contrast, TRIM5α and other restriction factors can dramatically change antiviral potency via single mutations at viral interaction interfaces (*Daugherty and Malik, 2012*; *Mitchell et al., 2012*). However, since the frequency of such gain-of-function mutations is unknown, it is unclear whether virus-binding surfaces in rapidly evolving antiviral factors are subject to the same evolutionary constraints as previously mapped for other proteins.

Here, we investigated the adaptive landscape of antiviral specificity conferred by the rapidly evolving, capsid-binding v1 loop of TRIM5α. To our surprise, we found that, rather than the evolutionary landscape of TRIM5α being narrowly constrained among all possible amino acid substitutions, the majority of random mutations in the v1 loop resulted in gains of antiviral restriction. We found that the primary v1 loop determinant for TRIM5α's restriction of HIV-1 and other lentiviruses is its net electrostatic charge. Furthermore, both rhesus and human TRIM5α proteins are highly resilient to mutation, in that they withstand more than half of all possible single amino acid mutations in the v1 loop without compromising their antiviral restriction abilities. This unexpectedly permissive landscape allows TRIM5α to sample a wide variety of mutations to maximize its chances of success in arms races with retroviruses.

## Results

### A deep mutational scan of the TRIM5α v1 loop

Despite their rapid evolution, primate TRIM5α orthologs have sampled relatively limited amino acid diversity at rapidly evolving positions within the capsid-binding v1 loop (*Figure 1A*). For example, although single amino acid changes at residue 332 are responsible for dramatic differences in antiviral restriction (*Li et al., 2006*), this residue repeatedly toggles between just three amino acids. The limited diversity is not due to evolutionary inaccessibility, since most amino acids that can be sampled with single nucleotide changes are not observed among primate TRIM5α orthologs (*Figure 1B*). There are two alternative explanations for this restricted diversity. First, it might suggest that adaptive gain-of-function mutations in TRIM5α are rare, with TRIM5α's evolutionary landscape mainly consisting of fitness valleys with only a few mutational avenues to reach fitness peaks (*Figure 1C*). Conversely, the limited diversity might be a consequence of epistatic interactions with other sites that constrain amino acid sampling, or even simple chance. Under this scenario, TRIM5α's evolutionary landscape may consist of numerous, wide peaks that tolerate substantial mutational variation (*Figure 1D*). We sought to differentiate between these possibilities by experimentally defining the evolutionary landscape of antiviral restriction over all possible single mutational steps in the v1 loop of both human and rhesus TRIM5α.

We took a deep mutational scanning (DMS) approach (*Fowler et al., 2010*) to measure the effect on antiviral restriction of all v1 loop single mutations in a pooled assay. We first generated a library of all single amino acid variants (including stop codons) within the rapidly evolving portion of the v1 loop (amino acids 330 to 340, *Figure 1A*), with a library diversity of 231 amino acid (352 nucleotide) variants (*Figure 2A*). The resulting TRIM5α variants were stably expressed via transduction into CRFK (cat renal fibroblast) cells, which naturally lack TRIM5α (*McEwan et al., 2009*). We transduced CRFK cells at a low dose to limit the integration of multiple variants into individual cells, thus generating a pool of cells each expressing a single TRIM5α point mutant. Libraries were represented with at least 500-fold coverage through all experimental steps to avoid bottlenecking library diversity.

Human TRIM5α only weakly restricts HIV-1 (*Jimenez-Guardeño et al., 2019*; *OhAinle et al., 2018*). However, single amino acid mutations in the v1 loop can substantially increase restriction

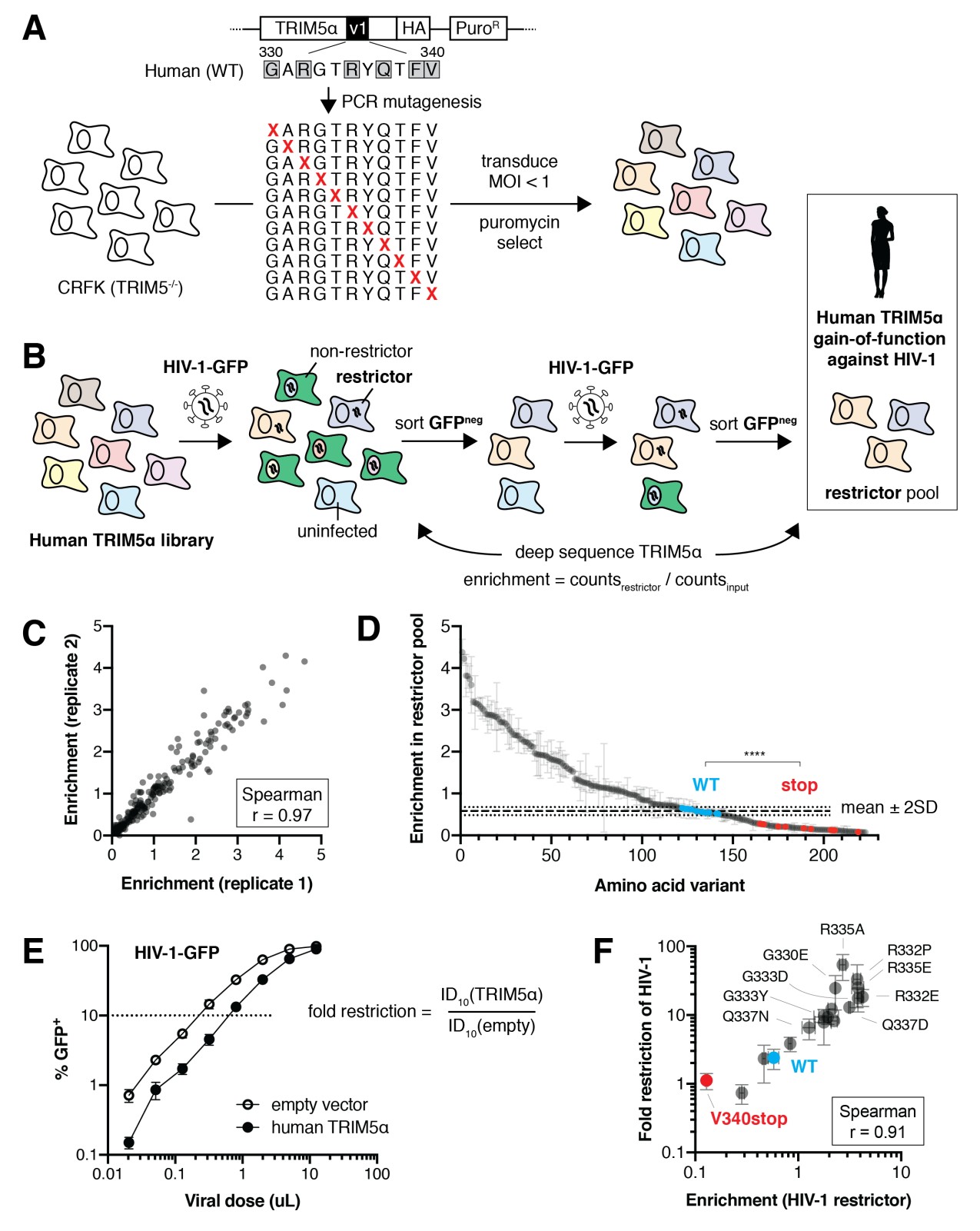

**Figure 2.** Selection scheme to identify human TRIM5α variants that gain HIV-1 restriction. (**A**) A DMS library, encoding all single amino acid variants within the v1 loop (rapidly evolving sites are boxed), was generated by PCR with degenerate NNS codons. The library was transduced into naturally TRIM5α-deficient CRFK cells at low MOI (multiplicity of infection) and selected using puromycin. Colors represent different TRIM5α variants. (**B**) Pooled TRIM5α-expressing cells were infected with HIV-1-GFP virus-like particles (VLPs) at a high dose. GFP-negative cells were FACS sorted, re-infected, and

*Figure 2 continued on next page*

*Figure 2 continued*

re-sorted. Restrictive TRIM5α variants were then sequenced, and variant frequencies were normalized to input representation. (C) Amino acid enrichment scores are highly correlated across two biological replicates. Each dot represents a unique amino acid sequence, averaged across synonymous codons. (D) Nonsense variants (red, n = 11) are depleted relative to WT (blue, n = 10) and most missense (gray) variants; ****p<0.0001, student's unpaired t-test. Enrichment is averaged across synonymous codons and replicates, except for WT variants, which are averaged only across replicates to better visualize variance (WT mean ±2 standard deviations is indicated). (E) HIV-1 fold-restriction by TRIM5α was measured by the increase in $ID_{10}$ (viral dose at which 10% of cells are infected) relative to an empty vector control. (F) Enrichment scores are highly correlated with HIV-1 restriction (n ≥ 3 biological replicates) for re-tested variants. (D-F) Error bars, SD.

The online version of this article includes the following source data for figure 2:

Source data 1. Quality control of DMS libraries.
Source data 2. Enrichment scores for human TRIM5α gain-of-function against HIV-1.
Source data 3. Validation of human TRIM5α gain-of-function screen against HIV-1.

---

activity (*Li et al., 2006*; *Pham et al., 2010*; *Pham et al., 2013*). To comprehensively assess how many single mutation variants of human TRIM5α had increased activity against HIV-1, we first performed a gain-of-function screen. We challenged the library of human TRIM5α variant-expressing cells with HIV-1 bearing a GFP reporter, at a dose infecting 98% of cells (*Figure 2B*). Because GFP expression becomes detectable only after integration of the HIV-1 proviral genome, cells expressing TRIM5α variants that restrict HIV-1 infection remain GFP-negative. However, ~2% of cells that were uninfected by chance would also be GFP-negative. Therefore, to enrich for cells expressing *bona fide* restrictive TRIM5α variants, we sorted the GFP-negative cells from the first round of infection and subjected them to a second round of HIV-1-GFP infection and sorting. Following this second round of selection, we deep sequenced the TRIM5α variants in the GFP-negative cell population. We normalized the count of each variant to its representation in the pre-selection cell population to determine its enrichment score, which should reflect the relative antiviral function of each TRIM5α variant.

Enrichment scores were highly correlated between two independent biological replicates (*Figure 2C*, Spearman r = 0.97). Furthermore, mutants bearing premature stop codons, which should be non-functional and depleted from the restrictor pool, were all among the most depleted variants (*Figure 2D*, red). Despite the weak (~2 fold) restriction of HIV-1 by wildtype (WT) human TRIM5α, variants containing synonymous nucleotide changes (no amino acid changes compared to WT, blue) had significantly higher enrichment scores than those containing stop codons (p<0.0001, student's unpaired t-test with Welch's correction), confirming that the assay worked as expected.

To investigate whether enrichment scores were truly representative of increased antiviral function, and to validate some of the novel amino acid changes that appeared to result in increased restriction, we made 16 targeted missense mutants from across the enrichment spectrum and challenged them individually with HIV-1-GFP. We determined their fold-restriction by determining the relative viral dose required to infect 10% of cells ($ID_{10}$) expressing a TRIM5α variant compared to an empty vector control; a larger viral dose is required to overcome TRIM5α-mediated restriction (*Figure 2E*). We confirmed that several previously described gain-of-function variants (*Li et al., 2006*; *Pham et al., 2010*; *Pham et al., 2013*) had increased antiviral activity and were highly enriched (*Figure 2F*: G330E, R332P, R332E, R335E). Moreover, we identified novel amino acid mutations that significantly increased antiviral activity, such as R335A and G333D, whereas moderately enriched variants (e.g. G333Y, Q337N) had correspondingly modest gains in HIV-1 restriction. Indeed, enrichment scores and fold-restriction were highly correlated across all mutants tested (Spearman r = 0.90). Thus, enrichment scores accurately reflect antiviral activity, validating our approach to simultaneously identify all single mutants with increased HIV-1 restriction. Therefore, in subsequent analyses, we use enrichment scores as a proxy for the antiviral restriction activity of TRIM5α mutants.

## Most single mutations in the v1 loop improve human TRIM5α restriction of HIV-1

Based on the limited amino acid diversity among primate TRIM5α v1 loops (*Figure 1A–B*), we expected that our DMS assay would reveal only a few beneficial mutations that improve human TRIM5α restriction of HIV-1. Contrary to this expectation, we found that more than half of all

missense variants (115, 57%) had enrichment scores that fell more than two standard deviations above WT TRIM5α (*Figure 2D*). Even if we limited our analysis to amino acid variants that are evolutionarily accessible via single-nucleotide changes from the WT *TRIM5α* sequence, this ratio did not change substantially (32, 54%). These enrichment scores represent dramatic gains in HIV-1 restriction, with the most potent variants (R332P and R335A) improving HIV-1 restriction ~15 fold relative to WT (33- and 39-fold restriction, respectively). Our findings indicate that the fitness landscape of the v1 loop is not narrowly constrained, but rather is remarkably permissive (*Figure 1D*), in that most single amino acid changes not seen in natural sequences enhance the ability of human TRIM5α to restrict HIV-1. Thus, TRIM5α has the capacity to readily evolve antiviral potency against HIV-1 via single mutations.

We analyzed whether a common biochemical mechanism could explain the unexpectedly high fraction of restrictive TRIM5α variants. We found that increased expression levels could explain some of the improvement in HIV-1 restriction, although several mutations (G333D, G333Y) improved restriction without increasing expression (*Figure 3—figure supplement 1A–B*). In contrast, most gains in HIV-1 restriction could be completely accounted for by a reduction in the electrostatic charge of the v1 loop (*Figure 3A*), regardless of expression level. Indeed, reducing the electrostatic charge always improved HIV-1 restriction, but did not always increase TRIM5α expression level (*Figure 3—figure supplement 1C–D*). Among all variants tested, mutation of the positively-charged residues 332 or 335 from the WT arginine (R) to any amino acid except lysine (K) significantly improved HIV-1 restriction (*Figure 3A–B*), consistent with previous reports on R332 variants (*Li et al., 2006*). Mutation of uncharged sites to K or R decreased TRIM5α restriction of HIV-1, whereas introducing a negatively charged aspartic acid (D) or glutamic acid (E) significantly increased HIV-1 restriction (*Figure 3C*). Since our DMS assay tests only one mutation at a time, all mutations to D or E occur in the context of at least one proximal positively charged residue. Therefore, we infer that the position-independent benefit of introducing D or E derives from offsetting pre-existing positive charge in the v1 loop that is detrimental to HIV-1 restriction. Indeed, reducing the net charge of the v1 loop explains all the highest enrichment scores (*Figure 3D*). Thus, we conclude that positive charge in the v1 loop is the dominant impediment to HIV-1 restriction by human TRIM5α.

Removal of positive charge, however, could not explain all of the improved HIV-1 restriction we observed. For example, despite its strict conservation in primate TRIM5α (*Figure 1A*), a glycine (G) at residue 333 compromises HIV-1 restriction. Mutation of G333 to most other amino acids significantly improves TRIM5α activity (*Figure 3B*). We confirmed this finding for several individual variants (*Figure 2F*: G333Y, G333D). We found a similar pattern for residue F339, which is disfavored for HIV-1 restriction, albeit not to the same extent as G333. Contrary to our initial expectations, there is only a weak association between rapidly evolving residues and residues whose mutation can significantly improve HIV-1 restriction: missense mutations in three of six rapidly evolving sites, versus one of five conserved sites, significantly improve HIV-1 restriction (*Figure 3B*). This result suggests that conserved positions in the vicinity of rapidly evolving sites possess unexpected potential to improve antiviral potency.

We also tested whether beneficial mutations might have additive effects on HIV-1 restriction by human TRIM5α. We combined several gain-of-function variants with the R332P mutation, previously described to potently restrict HIV-1 (*Yap et al., 2005*). However, we found that no double mutants tested increased HIV-1 restriction beyond that of R332P alone (*Figure 3E*). Instead, combination of one gain-of-function variant (R335E) with R332P resulted in loss of protein expression and HIV-1 restriction. Previous reports also found that most beneficial mutations are either non-additive or interfering, identifying only one combination (R332G with R335G) that was partially additive (*Li et al., 2006*; *Pham et al., 2010*; *Pham et al., 2013*). Given that R332P is one of the strongest gain-of-function variants we identified, it remains possible that other, more modest gain-of-function variants might additively improve HIV-1 restriction. Nevertheless, these results suggest that single gain-of-function mutations, such as R332P, can confer most or all the increased HIV-1 restriction potential onto human TRIM5α. Thus, remarkably, human TRIM5α appears to be located only one mutational step away from fitness peaks in its evolutionary landscape of potential adaptation against HIV-1.

Finally, we investigated whether gain-of-function mutations for HIV-1 restriction also conferred protection against other lentiviruses. We focused on lentiviruses whose restriction is v1 loop-dependent: either the entire v1 loop (*Figure 4—figure supplement 1*) or the R332P mutation from rhesus

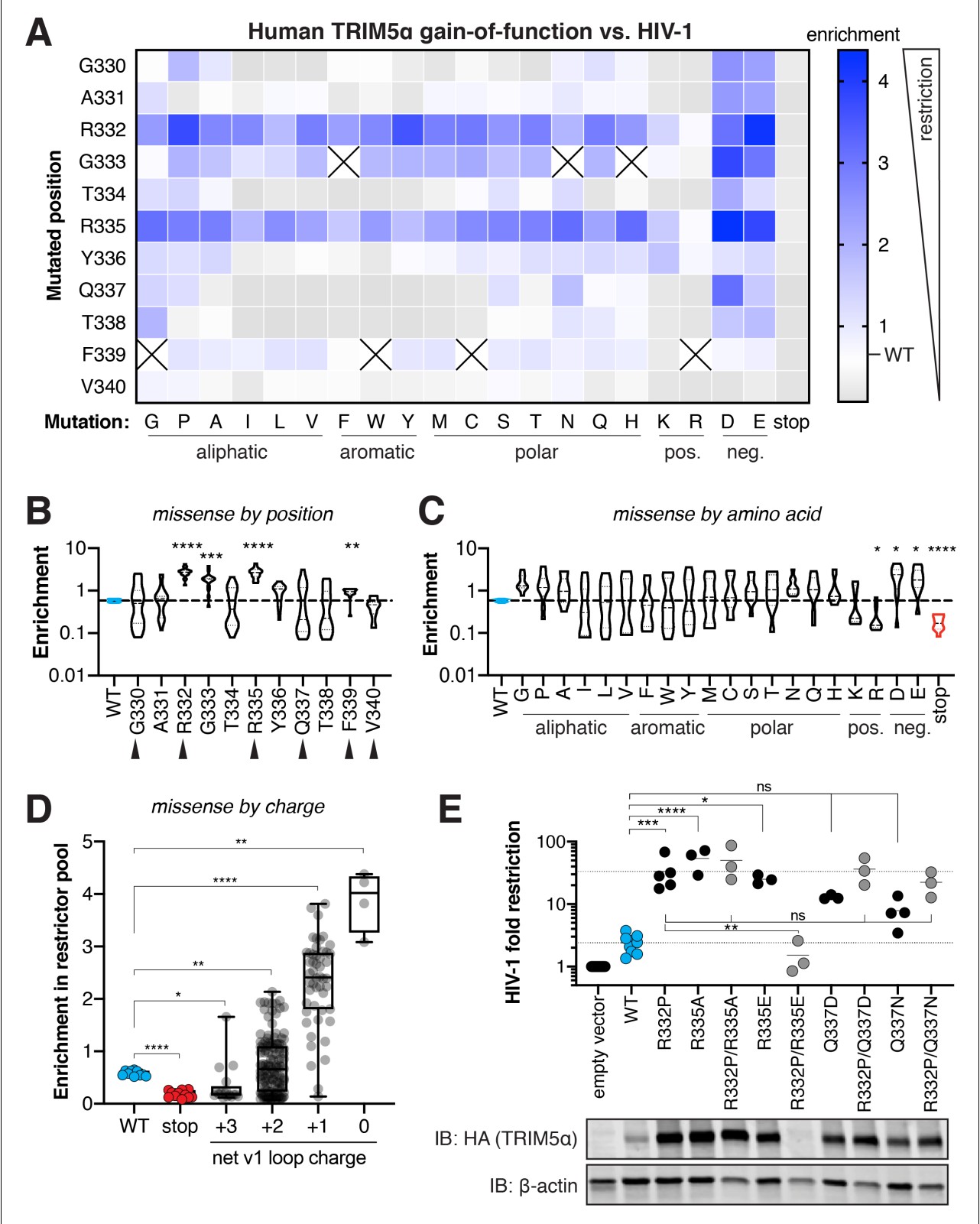

**Figure 3.** Many single mutations improve human TRIM5α restriction of HIV-1, primarily by removal of positive charge. (**A**) Enrichment in the HIV-1 restrictor pool relative to WT (white) for each TRIM5α variant, arrayed by position mutated and amino acid mutation, is indicated by color intensity. Variants marked with X were excluded due to low input representation. (**B-C**) Enrichment scores for each position across all amino acid variants (**B**) or each amino acid across all positions (**C**); statistics reported in comparison to WT. Rapidly evolving sites are indicated by black arrows. (**D**) Box plot of

*Figure 3 continued on next page*

*Figure 3 continued*

missense mutations grouped by their effect on the net v1 loop charge; WT has a net v1 charge of +2. (**E**) Gain-of-function mutations were tested against HIV-1 individually or in combination with R332P, and fold-restriction was determined as in *Figure 2E*. TRIM5α expression levels in CRFK cells were analyzed by immunoblot (IB) against the C-terminal HA tag (blot representative of three independent experiments). (**B-E**) *p<0.05, **p<0.01, ***p<0.001, ****p<0.0001; one-way ANOVA with Holm-Sidak's correction for multiple comparisons and (**B-D**) correction for unequal variances.

The online version of this article includes the following source data and figure supplement(s) for figure 3:

**Source data 1.** Enrichment scores and p-values for human TRIM5α gain-of-function against HIV-1.
**Source data 2.** Raw western blots of human TRIM5α variants.
**Source data 3.** Quantification of human TRIM5α western blots.
**Source data 4.** HIV-1 fold restriction by human TRIM5α double mutants.
**Figure supplement 1.** Some, but not all, human TRIM5α gain-of-function mutations against HIV-1 increase TRIM5α expression level.

TRIM5α (*Stremlau et al., 2005*) could confer human TRIM5α with substantial antiviral function. In each case, WT human TRIM5α only weakly restricts these lentiviruses (*Figure 4A*). However, the charge-altering mutations R332P and R335A increased restriction of all lentiviruses we tested, including HIV-2, SIVcpz (SIV infecting chimpanzees), and SIVmac (SIV infecting rhesus macaques). Introduction of negative charge (R332E, R335E, G330E, G333D, and G337D) also selectively improved restriction of HIV-1, SIVcpz, and HIV-2 but not SIVmac. Thus, positive charge at positions 332 and 335 appears to be generally detrimental for lentiviral restriction. Furthermore, although TRIM5α fitness landscapes are lentivirus-specific, many of the mutations we tested increased restriction against other lentiviruses. These data suggest that the evolutionary landscape for lentiviral restriction by TRIM5α is likely to be generally permissive, as it is for HIV-1.

## TRIM5α restriction of HIV-1 is resilient to single mutations

Our data show that novel antiviral potency is readily attainable by single amino acid changes in human TRIM5α (*Figures 2D* and *4A*). However, these gains might be just as easily lost through further mutation, since rapidly evolving antiviral proteins like TRIM5α continually adapt in their arms race with viruses. Therefore, in order to test whether newly acquired antiviral potency is fragile or resistant to mutation, we investigated the mutational resilience of the R332P variant of human TRIM5α, which inhibits HIV-1 ~15 fold more than WT (*Figure 2F*). To do so, we generated a v1 DMS library of human TRIM5α with R332P fixed in all variants. We challenged this pooled cell library with HIV-1-GFP, at a viral titer which human TRIM5α-R332P restricts to ~1% infection. In this case, we sorted and deep sequenced GFP-positive cells, so that enrichment (relative to initial representation) now reflects the degree to which each TRIM5α-R332P variant lost its antiviral function against HIV-1 (*Figure 4B*). As expected, we observed strong enrichment of stop codons in the non-restrictor pool and good correlation between biological replicates (*Figure 4C*).

Addition of positive charge by mutations to K or R at most positions in the v1 loop reduced HIV-1 restriction (*Figure 4—figure supplement 2A–B*). This preference against positive charge mirrors that of WT human TRIM5α, for which removal of positive charge increased HIV-1 restriction (*Figure 4—figure supplement 2C*). However, we found no other consistent biochemical constraints for HIV-1 restriction by TRIM5α-R332P (*Figure 4—figure supplement 2A–B*). Indeed, we found that the majority of missense variants (65%) did not weaken HIV-1 restriction by the R332P variant of human TRIM5α (*Figure 4C*). Thus, WT human TRIM5α is only one mutational step away from a fitness peak (*Figure 3*) that, once achieved, also exhibits a surprising degree of resilience to mutation. This implies that gains of restriction by TRIM5α are not likely to be compromised by its continued adaptation.

To determine if HIV-1 restriction is also resilient to mutation in a naturally occurring TRIM5α variant, we next assessed the likelihood that random mutations disrupt viral restriction by WT rhesus macaque TRIM5α, which strongly restricts HIV-1 in a manner that strictly requires the v1 loop (*Sawyer et al., 2005*). Like with human TRIM5α, we constructed a library of cells each expressing a single rhesus TRIM5α variant, with each variant containing a single mutation in the v1 loop (note that the v1 loop is slightly longer in macaques compared to humans, *Figure 1A*). We then challenged this pool of cells with HIV-1-GFP and sorted GFP-positive cells for subsequent deep sequencing. Rhesus TRIM5α variants containing premature stop codons were strongly enriched in the non-restrictor pool, whereas WT variants were significantly depleted (*Figure 5A*). In contrast, half of all

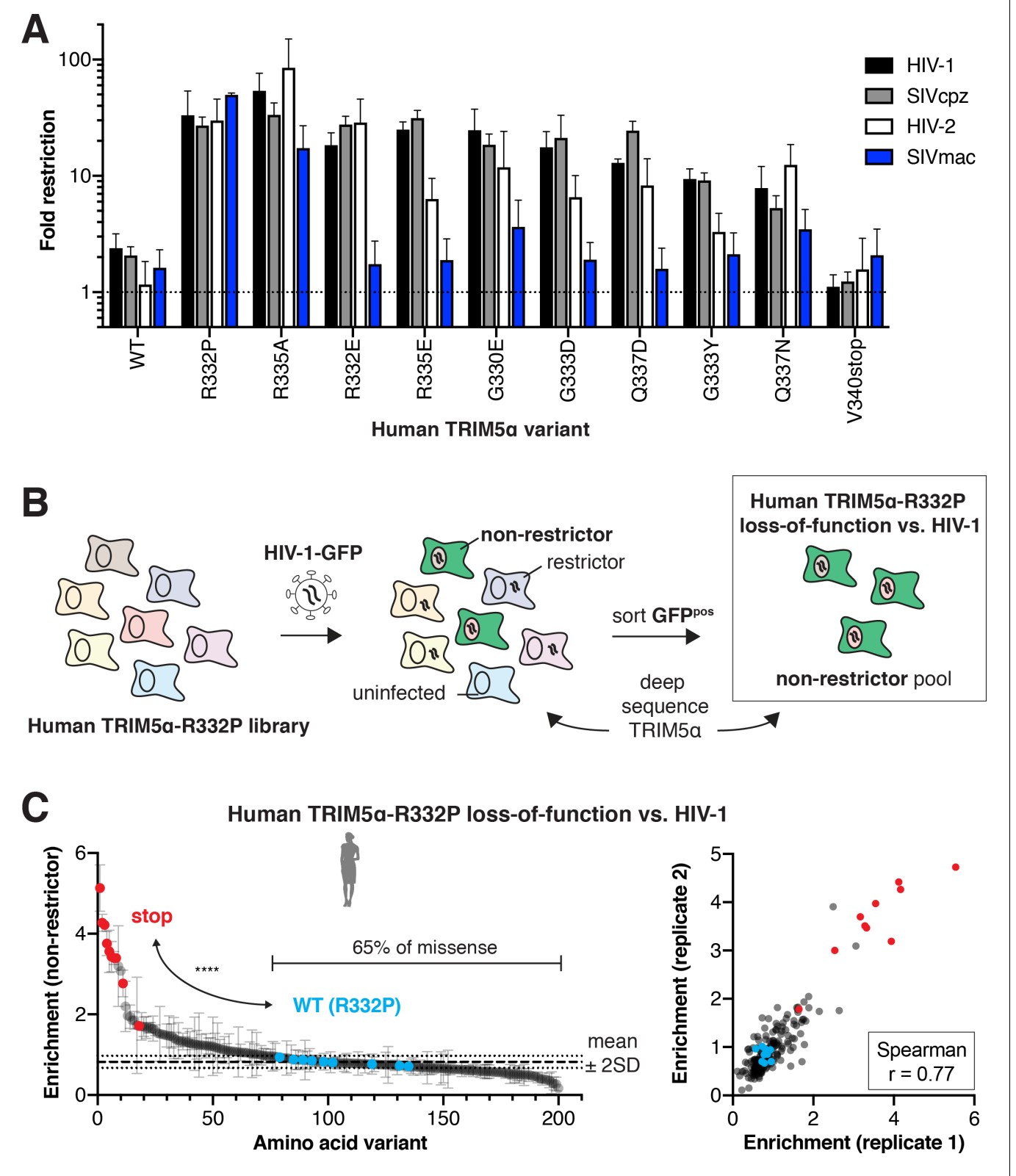

**Figure 4.** Evolutionary landscapes are generally permissive for evolving novel lentiviral restriction, which is resilient to most mutations once achieved. (A) CRFK cells expressing the indicated human TRIM5α variant were challenged with GFP-marked lentiviral VLPs to determine fold-restriction as in *Figure 2E*. Results from at least three independent experiments. (B) To determine whether newly acquired viral restriction tolerates mutations, a second human TRIM5α v1 DMS library was generated with R332P fixed in all variants. This library of cells was infected with HIV-1-GFP VLPs, and GFP-positive

*Figure 4 continued on next page*

*Figure 4 continued*

(non-restrictor) cells were sorted and sequenced. (**C**) Stop codon variants (red, n = 10) are highly enriched in the non-restrictor pool compared to WT (R332P, blue, n = 9) variants (****p<0.0001, student's unpaired t-test with Welch's correction), while 65% of all missense variants fall less than 2 SD above WT (R332P) mean. Enrichment scores between two biological replicates are well correlated. (**A, C**) Error bars, SD.

The online version of this article includes the following source data and figure supplement(s) for figure 4:

**Source data 1.** TRIM5α v1-dependence of lentivirus restriction.
**Source data 2.** Lentiviral restriction by human TRIM5α single mutants.
**Source data 3.** Enrichment scores and p-values for human TRIM5α-R332P loss-of-function against HIV-1.
**Figure supplement 1.** Lentiviral restriction by TRIM5α is v1-dependent.
**Figure supplement 2.** Biochemical preferences for human TRIM5α-R332P restriction of HIV-1.

missense mutations (125, 51%) fell within two standard deviations of WT. Even missense mutations accessible by single-nucleotide changes reflected this pattern (40, 55%).

By re-testing individual variants, we confirmed that enrichment scores negatively correlate with antiviral potency (*Figure 5B*). We tested seven variants enriched for loss-of-restriction (more than two standard deviations above WT) and found that six lost HIV-1 restriction. The seventh variant (L337N) was only slightly outside the two standard-deviation cut-off for enrichment and correspondingly only slightly worse than WT in terms of HIV-1 inhibition. Thus, enrichment in the non-restrictor pool represents *bona fide* loss of restriction. All the rhesus TRIM5α variants we report here represent novel loss-of-function mutations. Their loss of HIV-1 restriction cannot be explained by loss of expression or protein stability (*Figure 5C*). For example, the F340D, P341I, and P341G variants were all expressed at WT levels but lost HIV-1 restriction. Moreover, the T344E variant retained restriction despite reduced expression levels.

We also re-tested ten rhesus TRIM5α variants not significantly enriched for loss-of-restriction (*Figure 5B*). Two variants (P334M, G335I) enriched one standard deviation above WT correspondingly retained only partial HIV-1 restriction relative to WT rhesus TRIM5α. The eight remaining variants retained HIV-1 inhibition, consistent with their lack of enrichment relative to WT. Based on this validation, we conclude that roughly half of all v1 loop single point mutations do not significantly reduce HIV-1 restriction by rhesus TRIM5α. Thus, a natural rhesus TRIM5α antiviral variant, much like the human TRIM5α-R332P variant, displays considerable mutational resilience.

We expected that conserved residues should be less tolerant of changes than rapidly evolving sites. However, we found that mutations in only three of seven conserved sites, versus two of six rapidly evolving sites, led to significant loss of function (*Figure 5D*). Collectively, these results indicate that rhesus TRIM5α restriction of HIV-1 is highly robust to changes within the critical v1 loop at both rapidly evolving and conserved sites. The biochemical preferences for HIV-1 restriction are similar but not identical between rhesus and human TRIM5α. In both cases, the introduction of positive charge, particularly R, weakened HIV-1 inhibition (*Figure 5D*, compare to *Figure 3C*). In contrast, the introduction of bulky hydrophobic residues, including leucine (L), phenylalanine (F), and tryptophan (W), significantly impaired HIV-1 restriction by rhesus TRIM5α but did not affect the potency of human TRIM5α. These data suggest that both universal as well as lineage-specific requirements for the v1 loop shape TRIM5α restriction of HIV-1.

Our findings with TRIM5α restriction of HIV-1 suggest that single mutations can readily achieve gain-of-function. In contrast, loss-of-function mutations are not so abundant as to make adaptation unlikely. Thus, the evolutionary landscape of TRIM5α appears to resemble 'rolling hills' (*Figure 1D*) rather than rare, sharp peaks (*Figure 1C*).

## Resilience of antiviral restriction is a general property of TRIM5α adaptation

Our DMS analyses of human TRIM5α revealed unexpected ease of gaining antiviral potency against HIV-1 and potentially other lentiviruses. However, gains in potency against one virus might be offset by a concomitant loss of function against other viruses, as previously seen for the antiviral protein MxA (*Colón-Thillet et al., 2019*). Such functional tradeoffs might partially explain the evolutionary constraints acting on primate TRIM5α sequences. To explore this possibility, we investigated the mutational resilience of N-tropic murine leukemia virus (N-MLV) restriction by TRIM5α. N-MLV is strongly inhibited by both rhesus and human TRIM5α, and this activity is at least partly dependent

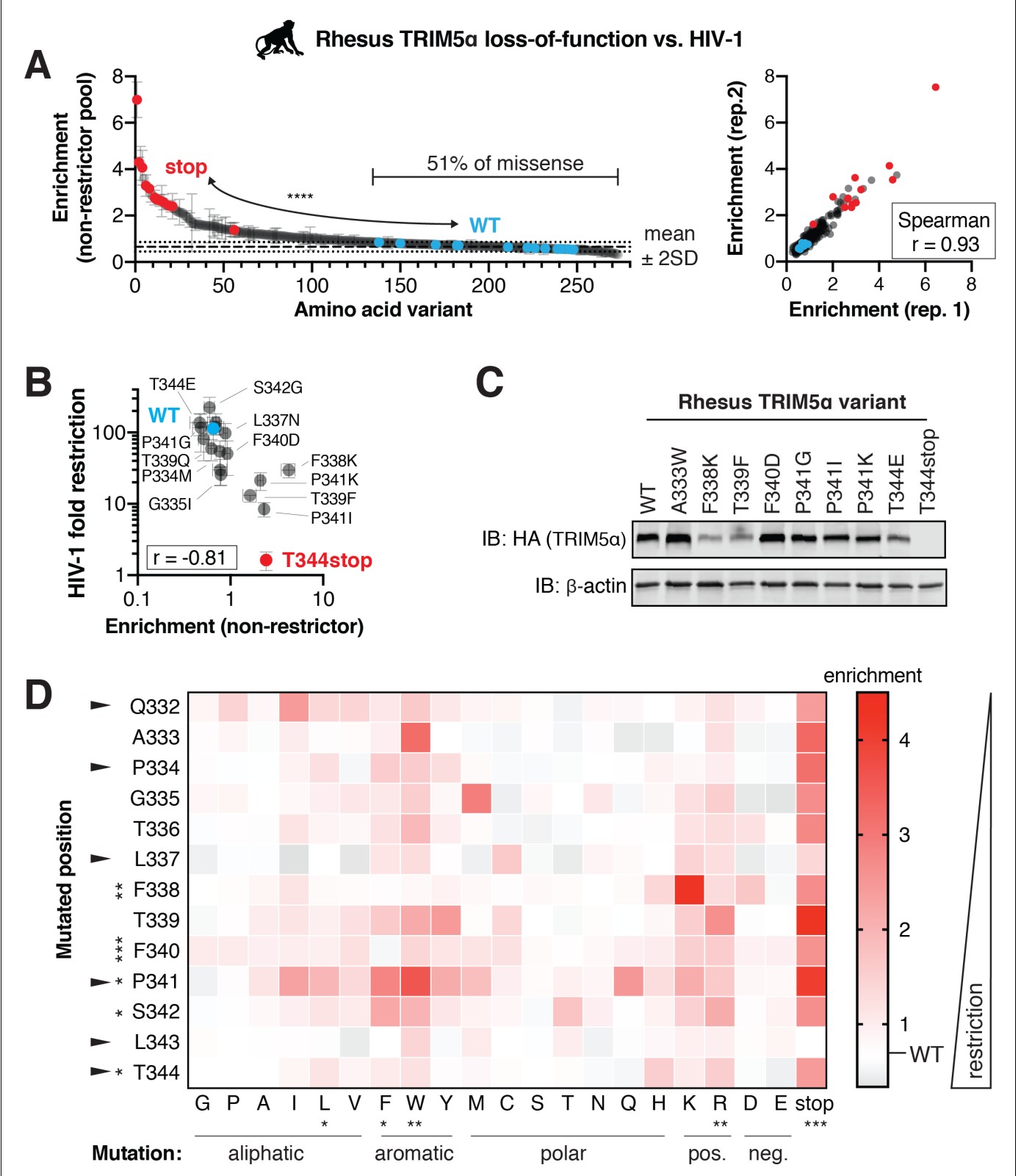

**Figure 5.** Rhesus macaque TRIM5α restriction of HIV-1 tolerates many mutations. A rhesus TRIM5α v1 DMS library was infected with HIV-1-GFP VLPs, and GFP-positive (non-restrictor) cells were sorted and sequenced. (A) Nonsense variants are highly enriched in the non-restrictor pool compared to WT (n = 13; ****p<0.0001, student's unpaired t-test with Welch's correction), while half of all missense variants fall less than 2 SD above WT. Enrichment scores between two biological replicates are highly correlated. (B) Re-testing individual variants confirms that enriched variants have partially lost HIV-1

*Figure 5 continued on next page*

*Figure 5 continued*

restriction, while depleted variants do not differ from WT. Spearman r; error bars, SD; n ≥ 3 biological replicates. (C) Steady-state levels of TRIM5α variants stably expressed in CRFK cells; results representative of two independent immunoblots (IB). (D) Enrichment in the HIV-1 non-restrictor pool relative to WT (white) for each variant, arrayed by position and amino acid mutation, is indicated by color intensity. The color scale was slightly compressed to avoid exaggerating a single mutant (L343stop) with enrichment >4.5. Rapidly evolving sites are indicated with arrows. Statistical tests compare each position (across all variants) or each amino acid (across all positions) to WT, one-way ANOVA with Geisser-Greenhouse non-sphericity and Holm-Sidak's multiple comparisons corrections; *p<0.05, **p<0.01, ***p<0.001, ****p<0.0001.

The online version of this article includes the following source data for figure 5:

**Source data 1.** Enrichment scores and p-values for rhesus TRIM5α loss-of-function against HIV-1.
**Source data 2.** Validation of rhesus TRIM5α loss-of-function screen against HIV-1.
**Source data 3.** Raw western blots of rhesus TRIM5α.

on the v1 loop (*Ohkura et al., 2006*; *Perron et al., 2006*). We infected cells expressing either the rhesus (*Figure 6A*) or WT human TRIM5α (*Figure 6B*) v1 DMS libraries with GFP-marked N-MLV, sorted GFP-positive cells, and sequenced the non-restrictor variants. For both selections, stop codon variants were significantly more enriched than WT variants in the non-restrictor pool.

Similar to HIV-1 restriction, we found that most missense mutations (143, 58%) in rhesus TRIM5α were tolerated for N-MLV restriction (*Figure 6A*). However, some missense mutations dramatically reduced N-MLV restriction, affirming that the v1 loop is indeed critical for inhibition of N-MLV (*Figure 6—figure supplement 1A–C*). In particular, hydrophobic and especially aromatic residues at most positions in the v1 loop significantly decreased N-MLV restriction. This preference against aromatic residues is similar between HIV-1 and N-MLV restriction. However, N-MLV restriction is insensitive to the introduction of positively charged residues, which disrupt HIV-1 inhibition (*Figure 6—figure supplement 1D*). These results indicate that the evolutionary landscape for rhesus TRIM5α against N-MLV is distinct from that of HIV-1. Nevertheless, the overall degree of mutational resilience against both viruses is remarkably similar: less than half of all missense mutations disrupt restriction of either virus.

Human TRIM5α restriction of N-MLV was even more resilient to mutation than rhesus TRIM5α. Almost all variants (187, 92%) had no effect on N-MLV restriction (*Figure 6B*, *Figure 6—figure supplement 2*). Indeed, our selection for non-restrictive variants only strongly enriched for stop codons. This extreme mutational resilience may reflect the massive potency (>250 fold restriction, data not shown) of human TRIM5α against N-MLV, and/or a decreased reliance on the v1 loop for N-MLV recognition by human TRIM5α (*Perron et al., 2006*). We validated several human TRIM5α mutants as retaining nearly WT levels of N-MLV restriction (*Figure 6—figure supplement 2C*). Thus, both rhesus and human TRIM5α inhibition of N-MLV is highly resistant to mutations, allowing mutational flexibility without loss of pre-existing antiviral restriction. These results, in conjunction with the substitution tolerance of HIV-1 restriction by human R332P and WT rhesus TRIM5α, indicate that mutational resilience is a general property of TRIM5α's rapidly evolving v1 loop.

## Discussion

Antiviral restriction factors are locked in high-stakes tit-for-tat evolutionary arms races with target viruses. However, viruses would appear to have the upper hand in these battles because of their higher mutation rates, shorter generation times, and larger population sizes. Although host genomes have the advantage of encoding a diverse, polygenic immune response, evolutionary constraints acting on innate immune genes could curtail their adaptive potential. Here, using deep-mutational scanning approaches combined with viral infection assays, we investigated the evolutionary landscape of adaptation of the most rapidly evolving segment, the disordered v1 loop, of the retroviral restriction factor TRIM5α. We focused on this loop because of its critical role in adapting to changing viral repertoires.

We found two attributes of this evolutionary landscape that favor host immune evolution. First, human TRIM5α readily gains significant HIV-1 restriction: roughly half of all single missense mutations allow human TRIM5α to better restrict HIV-1 (*Figures 2–3*). Based on our results, we infer that positive charge is the dominant impediment to HIV-1 inhibition in human TRIM5α (*Figure 3D*).

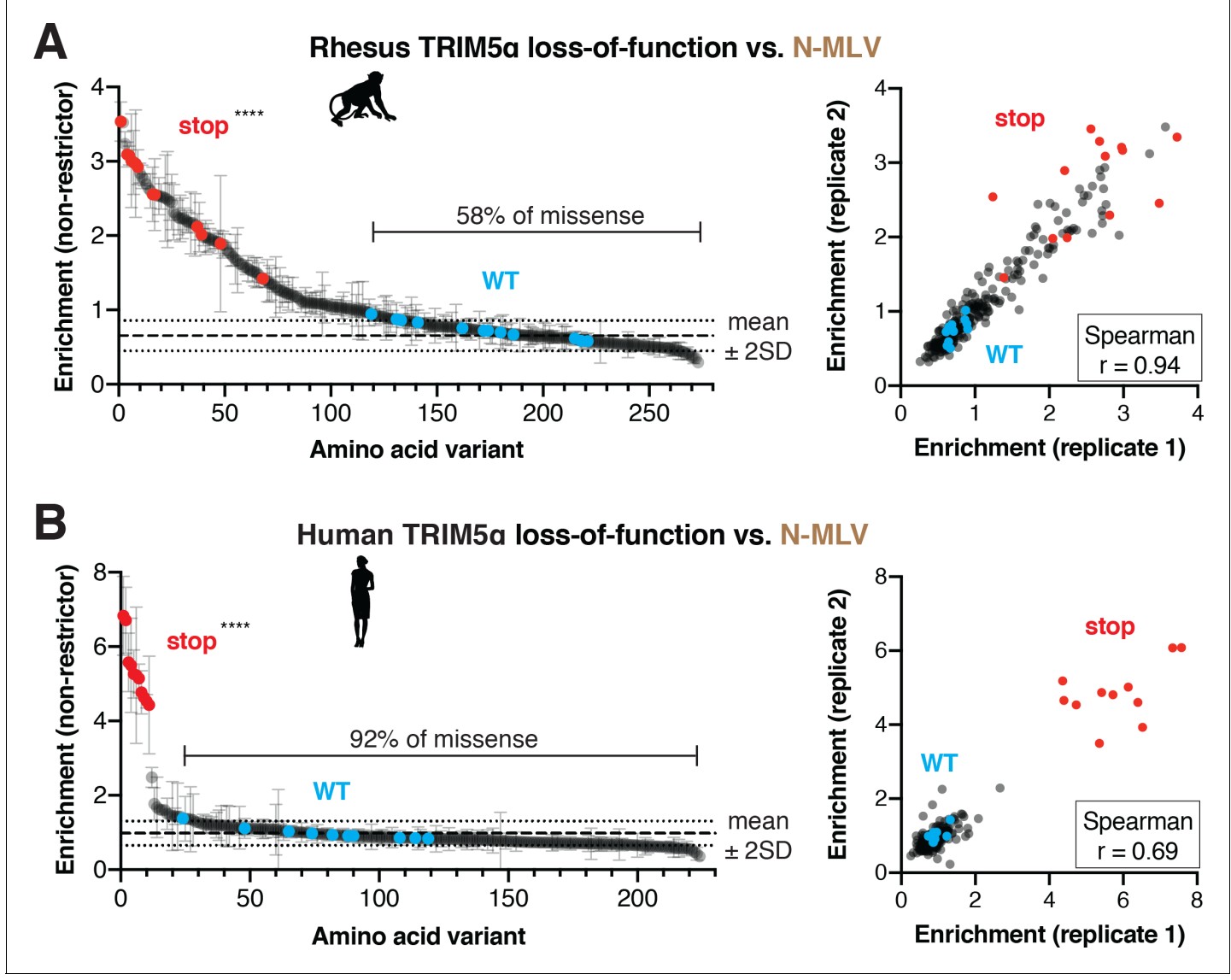

**Figure 6.** N-MLV restriction by primate TRIM5α is robust to single point mutations. The rhesus (**A**) or WT human (**B**) v1 DMS libraries were infected with N-MLV-GFP VLPs, and GFP-positive (non-restrictor) cells were sorted and sequenced. Nonsense variants are highly enriched in the non-restrictor pool compared to WT in both screens; ****p<0.0001, student's unpaired t-test with Welch's correction. Error bars, SD. The fraction of all missense variants less than 2SD above WT mean is indicated. Enrichment scores between two biological replicates are highly correlated.

The online version of this article includes the following source data and figure supplement(s) for figure 6:

**Source data 1.** Enrichment scores and p-values for human and rhesus TRIM5α loss-of-function against N-MLV.
**Source data 2.** Validation of human and rhesus TRIM5α loss-of-function screen against N-MLV.
**Figure supplement 1.** Biochemical preferences for rhesus TRIM5α restriction of N-MLV are distinct from HIV-1.
**Figure supplement 2.** Missense mutations do not disrupt human TRIM5α restriction of N-MLV.
**Figure supplement 3.** Summary of deep mutational scanning results.

Removal of this positive charge improved human TRIM5α restriction not only of HIV-1 but also of multiple lentiviruses (*Figure 4A*). Recent findings revealed that cyclophilin A (CypA) protects the HIV-1 capsid from TRIM5α recognition (*Kim et al., 2019*; *Selyutina et al., 2020*; *Veillette et al., 2013*). Although structural studies currently lack sufficient resolution to observe the molecular details of the TRIM5α–capsid interaction, we speculate that positive charge in the v1 loop impairs an interaction between TRIM5α and the capsid's CypA-binding site via electrostatic repulsion. Alternatively, the detrimental effect of positive charge might largely be accounted for by lowering TRIM5α

expression level (*Figure 3—figure supplement 1C–D*). Increasing human TRIM5α expression has been shown to improve HIV-1 restriction (*Richardson et al., 2014*), although we identified at least one mutation (V340H) that improved expression without increasing activity against HIV-1 and at least one mutation (G333D) that reduced positive charge and improved HIV-1 restriction without increasing expression level (*Figure 3—figure supplement 1B*).

Surprisingly, our comprehensive DMS analyses also revealed that both rapidly evolving and conserved residues can contribute to antiviral adaptation (*Figure 3B*). For instance, many variants at position 333 of human TRIM5α led to increased restriction of HIV-1 and other lentiviruses (*Figures 3A* and *4A*). Thus, it is unclear why simian primates have retained a glycine at this position (333 in human, 335 in macaques). One possibility is that changes in this residue might be generally deleterious for TRIM5α function, yet subsequent analyses revealed little or no impairment of TRIM5α antiviral functions (*Figure 5D*, *Figure 6—figure supplement 1A*, *Figure 6—figure supplement 2A*). More broadly, we identified many v1 loop mutations that improved human TRIM5α restriction of HIV-1, and even other lentiviruses, without impairing N-MLV antiviral function (*Figure 6—figure supplement 2D*). This permissivity stands in stark contrast to the natural evolution of TRIM5α, which has sampled relatively limited amino acid diversity (*Figure 1B*). The discrepancy might be explained by recurrent selection by a viral lineage distinct from the viruses we tested here (HIV-1 and N-MLV). For example, since the critical R332 was fixed at the common ancestor of humans, chimps, and bonobos ~7 million years ago, it is possible that R332 improved restriction of a paleovirus that has since gone extinct. Alternatively, TRIM5α's constrained natural evolution could reflect a cellular function independent of viral capsid recognition. For example, TRIM5α has been shown to induce some innate immune signaling even in the absence of infection (*Lascano et al., 2016*; *Pertel et al., 2011*; *Tareen and Emerman, 2011*). Many gain-of-antiviral-function mutations increased TRIM5α expression levels (*Figure 3—figure supplement 1*), and this increased expression might increase aberrant signaling by TRIM5α, driving chronic immune activation that can be costly for the host (*Ashley et al., 2012*; *Okin and Medzhitov, 2012*).

Our analyses uncovered a second unexpected, advantageous aspect of TRIM5α's evolutionary landscape: its antiviral restriction displays remarkable mutational resilience across multiple orthologs and against two divergent retroviruses. 51–92% of all possible missense variants retain antiviral activity (*Figure 6—figure supplement 3*). This resilience is manifest even when potent antiviral activity is newly acquired via a single mutation, as with the R332P variant of human TRIM5α against HIV-1. Therefore, we conclude that the fitness landscape of TRIM5α's rapidly evolving v1 loop resembles 'rolling hills' (*Figure 1D*), in which valleys are infrequent and only one evolutionary step removed from mutationally tolerant plateaus.

TRIM5α's permissive landscape contrasts with the relative inflexibility of ligand-binding domains in the core of evolutionarily conserved proteins (*Guo et al., 2004*; *McLaughlin et al., 2012*; *Suckow et al., 1996*). However, these studies found increased mutational tolerance in peripheral, disordered loops not involved in critical functions like ligand recognition, where mutations are less likely to disrupt protein structure. The use of flexible loops for viral ligand binding by TRIM5α, as well as MxA (*Mitchell et al., 2012*), thus grants rapidly evolving restriction factors mutational flexibility without significant risk of disrupting core protein structure. Nevertheless, the degree of mutational plasticity in TRIM5α's v1 loop is remarkable given that this loop is essential for ligand binding, suggesting that even functional constraints do not narrow its evolutionary landscape. Intriguingly, TRIM5α's reliance on the v1 loop for specificity mirrors that of antibodies' dependence on complementarity-defining loops for antigen recognition. Indeed, a high degree of mutational tolerance within complementarity-defining loops allows somatic hypermutation to significantly increase antibody-antigen affinity (*Daugherty et al., 2000*; *NISC Comparative Sequencing Program et al., 2017*).

Overall, our analyses reveal not only many paths for TRIM5α to gain antiviral function but also an unexpectedly low probability of losing antiviral function via single mutations. Such landscapes should be highly advantageous to host genomes in evolutionary arms races with viruses. Mutational tolerance allows the accumulation of neutral variants that do not compromise antiviral function among antiviral genes in a population. Many of these novel variants may carry the capacity to restrict additional viruses, whether these result from cross-species transmissions or mutations that allow species-matched viruses to evade recognition by the dominant antiviral allele. Indeed, mutational tolerance has been shown to facilitate the evolution of de novo functions through the accumulation of neutral

mutations (*Draghi et al., 2010*; *Hayden et al., 2011*). Although rare in human populations (*Clarke et al., 2017*), extensive polymorphism within the v1 loop of TRIM5α in Old World monkeys results in diverse antiviral repertoires that have been maintained by balancing selection (*Newman et al., 2006*). Thus, TRIM5α appears to evolve with low-cost, high-gain fitness landscapes that favor its success in co-evolutionary battles with rapidly evolving retroviruses.

# Materials and methods

Key resources table

| Reagent type (species) or resource | Designation | Source or reference | Identifiers | Additional information |
|---|---|---|---|---|
| Gene (*Homo sapiens*) | TRIM5α | NCBI | NM_033034.2 | |
| Gene (*Macaca mulatta*) | TRIM5α | NCBI | NM_001032910.1 | |
| Strain, strain background (*Escherichia coli*) | DH5α | NEB | C2987H | Chemically competent cells |
| Cell line (*Homo sapiens*) | HEK-293T/17 | ATCC | CRL-11268; RRID:CVCL_1926 | Purchased fresh stock from ATCC |
| Cell line (*Felis catus*) | CRFK | ATCC | CCL-94; RRID:CVCL_2426 | Purchased fresh stock from ATCC |
| Cell line (*Felis catus*) | CRFK + Human TRIM5α | This study | | Cell line stably expressing human TRIM5α (from ~ single random integration event per cell); see Materials and methods |
| Cell line (*Felis catus*) | CRFK + Human-v1DMS TRIM5α | This study | | Cell line stably expressing library of human TRIM5α single amino acid variants; see Materials and methods |
| Cell line (*Felis catus*) | CRFK + Human-R332P TRIM5α | This study | | Cell line stably expressing human TRIM5α-R332P; see Materials and methods |
| Cell line (*Felis catus*) | CRFK + Human-R332P -v1DMS TRIM5α | This study | | Cell line stably expressing library of human TRIM5α-R332P single amino acid variants; see Materials and methods |
| Cell line (*Felis catus*) | CRFK + Rhesus TRIM5α | This study | | Cell line stably expressing rhesus TRIM5α; see Materials and methods |
| Cell line (*Felis catus*) | CRFK + Rhesus-v1DMS TRIM5α | This study | | Cell line stably expressing library of rhesus TRIM5α single amino acid variants; see Materials and methods |
| Antibody | anti-HA.11 (mouse monoclonal) | Biolegend | 901516; RRID:AB_2565335 | (1:1000) |
| Antibody | anti-β-actin (rabbit polyclonal) | Abcam | ab8227; RRID:AB_2305186 | (1:5000) |
| Antibody | IRDye 680RD anti-mouse (donkey polyclonal) | LI-COR | 926–68072; RRID:AB_10953628 | (1:10,000) (secondary) |
| Antibody | IRDye 800CW anti-rabbit (donkey polyclonal) | LI-COR | 926–32213; RRID:AB_621848 | (1:10,000) (secondary) |
| Recombinant DNA reagent | MD2.G | Addgene | 12259 | VSV-G expression, CMV promoter |
| Recombinant DNA reagent | L-VSV-G | PMID:9245614 | | VSV-G expression, Tat-driven |

*Continued on next page*

*Continued*

| Reagent type (species) or resource | Designation | Source or reference | Identifiers | Additional information |
|---|---|---|---|---|
| Recombinant DNA reagent | CMV-Tat | PMID:9245614 | | Tat expression, CMV promoter |
| Recombinant DNA reagent | pHIV-ZsGreen | PMID:18371425 | | HIV-1 transfer vector |
| Recombinant DNA reagent | pALPS-eGFP | PMID:30546110 | | HIV-1 transfer vector |
| Recombinant DNA reagent | p8.9NdSB bGH Blp1 BstEII | PMID:15479815 | | NL4.3 HIV-1 gag/pol |
| Recombinant DNA reagent | p8.9NdSB bGH Blp1 BstEII HIV-2 CA | PMID:26181333 | | NL4.3 HIV-1 gag/pol with HIV-2$_{ROD}$ CA residues 1–202 |
| Recombinant DNA reagent | pHIV-MAC | PMID:18417575 | | HIV-1 gag/pol with SIVmac239 CA residues 1–146 (A77V) |
| Recombinant DNA reagent | pHIV-Gb2 | PMID:18417575 | | HIV-1 gag/pol with SIVcpzGab2 CA residues 1–146 |
| Recombinant DNA reagent | pCIG3N | PMID:10906195 | | N-MLV gag/pol |
| Recombinant DNA reagent | pQCXIP | Takara Bio | 631516 | Puromycin-selectable retroviral expression cassette |
| Recombinant DNA reagent | pQCXIP-eGFP | This study | | N-MLV transfer vector; see Materials and methods |
| Recombinant DNA reagent | pQCXIP-Human TRIM5α-HA | This study | | Stable expression of human TRIM5α; see Materials and methods |
| Recombinant DNA reagent | pQCXIP-Human-R332P TRIM5α-HA | This study | | Stable expression of human TRIM5α-R332P; see Materials and methods |
| Recombinant DNA reagent | pQCXIP-Human-v1$_{rhesus}$ TRIM5α-HA | This study | | Stable expression of human TRIM5α containing rhesus v1 loop; see Materials and methods |
| Recombinant DNA reagent | pQCXIP-Rhesus TRIM5α-HA | This study | | Stable expression of rhesus TRIM5α; see Materials and methods |
| Peptide, recombinant protein | Q5 high-fidelity DNA polymerase | NEB | M4091L | Used for Quikchange and DMS library construction |
| Commercial assay or kit | NEBuilder HiFi DNA assembly cloning kit | NEB | E5520S | Gibson assembly kit |
| Commercial assay or kit | PureYield plasmid miniprep kit | Promega | A2495 | Transfection-quality plasmid miniprep (low LPS) |
| Commercial assay or kit | NuceloBond Xtra midiprep kit | Takara Bio | 740410.50 | Transfection-quality plasmid midiprep (low LPS) |
| Commercial assay or kit | DNeasy Blood and Tissue kit | Qiagen | 69504 | Genomic DNA purification |
| Commercial assay or kit | QIAquick PCR purification kit | Qiagen | 28106 | PCR cleanup kit |
| Commercial assay or kit | Agencourt Ampure XP beads | Beckman Coulter | A63880 | Double-sided size selection of PCR products |

*Continued on next page*

*Continued*

| Reagent type (species) or resource | Designation | Source or reference | Identifiers | Additional information |
|---|---|---|---|---|
| Chemical compound, drug | Trans-IT 293T transfection reagent | Mirus Bio | MIR 2700 | Transfect plasmids into HEK-293T |
| Chemical compound, drug | polybrene | Sigma | TR-1003-G | Transduction reagent |
| Cchemical compound, drug | puromycin | Fisher | 50488918 | Antibiotic selection for stably transduced cells |
| Software, algorithm | DMS data analysis | This study | https://github.com/jtenthor/T5DMS_data_analysis | R scripts for DMS data analysis; see Materials and methods |
| Software, algorithm | PAML | PMID:967129 | | For analysis of rapid evolution |

## Plasmids and cloning

All virus-like particles (VLPs) were generated using three plasmids to ensure a single round of infectivity: a pseudotyping plasmid for transient expression of the VSV-G envelope protein (pMD2.G, Addgene plasmid #12259, gift from Didier Trono), a plasmid for transient expression of the viral gag/pol, and a transfer vector encoding a green fluorescent protein (GFP) integration reporter between the corresponding viral LTRs. HIV-1 VLPs were made with the transfer vector pHIV-ZsGreen (*Welm et al., 2008*); HIV-2 and SIV VLPs used the pALPS-eGFP transfer vector (*McCauley et al., 2018*); and N-MLV was made with pQCXIP-eGFP, encoding GFP between the EcoRI and ClaI sites of pQCXIP (P.S. Mitchell, unpublished). HIV-1 VLPs were made with p8.9NdSB bGH BlpI BstEII, encoding the NL4.3 HIV-1 gag/pol (*Berthoux et al., 2004*). HIV-2 VLPs used a chimeric gag/pol, in which the HIV-1 CA sequence (residues 1–202) was replaced by HIV-2$_{ROD}$ (p8.9NdSB bGH BlpI BstEII HIV-2 CA) (*Pizzato et al., 2015*). For SIV VLPs, pCRV1-based gag/pol chimeric vectors replaced the HIV-1 CA-NTD (residues 1–146) with the corresponding residues of either SIVmac239, a virus passaged in rhesus macaques, here SIVmac (pHIV-MAC, containing an A77V mutation); or SIVcpzGab2, a natural isolate from chimpanzee, here SIVcpz (pHIV-Gb2) (*Kratovac et al., 2008*). N-MLV VLPs were generated using pCIG3N, encoding the N-MLV gag/pol (*Bock et al., 2000*). For stable expression of TRIM5α constructs from pQCXIP, the MLV gag/pol was transiently expressed from JK3; the pseudotyping envelope protein was transiently expressed from the L-VSV-G plasmid and driven by expression of Tat from the CMV-Tat plasmid.

C-terminally HA-tagged human, human with the rhesus macaque v1 loop, and rhesus macaque TRIM5α (*Sawyer et al., 2005*) were amplified and cloned into pQCXIP (Takara Bio, Kusatsu, Shiga, Japan), just upstream of the IRES–puromycin resistance cassette, between the EcoRI and NotI restriction sites. Targeted TRIM5α mutations were generated by Quikchange PCR using primers containing the desired point mutation flanked by 17–25 nucleotides of homology on each side of the mutation. Primestar polymerase (Takara Bio) was used to minimize errors during full-plasmid amplification, followed by DpnI digestion of unmodified parent DNA. All plasmids were cloned into high-efficiency chemically competent DH5α (NEB, Ipswich, MA). Plasmids were purified using PureYield miniprep kits (Promega, Madison, WI), and coding sequences were verified by complete sequencing. See *Table 1* for all primers used in cloning and sequencing.

Deep mutational scanning libraries were generated using degenerate primers to amplify TRIM5α-HA in pQCXIP using high-fidelity Q5 polymerase (NEB). Degenerate primers contained a single NNS codon (N = A/T/C/G, S = C/G), which encodes all 20 amino acids with only one stop codon among 32 possibilities. For each of the 11 or 13 codons in the v1 loop of human or rhesus TRIM5α, respectively, the two halves of TRIM5α (on either side of randomized codon) were amplified separately with shared flanking primers and unique internal primers for each codon (*Table 1*). For the human TRIM5α library in which R332P was fixed, internal primers matched the R332P variant of TRIM5α and codon 332 was not randomized. Internal primers encoded NNS at the designated codon flanked by 17–25 nucleotides of homology on each side; the forward and reverse internal primers shared 17–25 nucleotides of homology with each other to promote hybridization between the N- and C-terminal PCR fragments. The codon-matched N- and C-terminal fragments were combined and amplified

**Table 1.** Primers used in this study.

| Primer | Use | Sequence |
|---|---|---|
| Subcloning TRIM5α constructs (human, human-v1rhesus, rhesus) into pQCXIP | | |
| oJT029 | Amplify human or rhesus TRIM5α-HA (Fwd), add NotI site, for cloning into pQCXIP | caagcggccgcgccaccATGGCTTCTGGAATC |
| oJT030 | Amplify human or rhesus TRIM5α-HA (Rev), add EcoRI site, for cloning into pQCXIP | gcggaattcTCAagcgtagtctgggacgtc |
| DMS library construction | | |
| oJT037 | Flanking primer for all DMS library construction (Fwd), amplifies pQCXIP backbone 5' of NotI site for Gibson cloning with pQCXIP-TRIM5α digested with NotI and BamHI | acctgcaggaattgatccgcggcc |
| oJT038 | Flanking primer for rhesus DMS library construction (Rev), amplifies rhesus TRIM5α 3' of BamHI site for Gibson cloning with pQCXIP-TRIM5α digested with NotI and BamHI | GGATTGGAAGCCAGCACATACCCCCAG |
| oJT003 | Randomize rhesus TRIM5α at codon Q332 (Fwd primer, use w/oJT038 for C-term half) | CGGAACCCACAGATAATGTAT NNSGCACCAGGGACATTATTTAC |
| oJT004 | Randomize rhesus TRIM5α at codon Q332 (Rev primer, use w/oJT037 for N-term half) | GTAAATAATGTCCCTGGTGCSN NATACATTATCTGTGGGTTCCG |
| oJT005 | Randomize rhesus TRIM5α at codon A333 (Fwd primer, use w/oJT038 for C-term half) | GAACCCACAGATAATGTATCAG NNSCCAGGGACATTATTTACGTTTC |
| oJT006 | Randomize rhesus TRIM5α at codon A333 (Rev primer, use w/oJT037 for N-term half) | GAAACGTAAATAATGTCCCTGG SNNCTGATACATTATCTGTGGGTTC |
| oJT007 | Randomize rhesus TRIM5α at codon P334 (Fwd primer, use w/oJT038 for C-term half) | CCACAGATAATGTATCAGGCAN NSGGGACATTATTTACGTTTCCG |
| oJT008 | Randomize rhesus TRIM5α at codon P334 (Rev primer, use w/oJT037 for N-term half) | CGGAAACGTAAATAATGTCCCSN NTGCCTGATACATTATCTGTGG |
| oJT009 | Randomize rhesus TRIM5α at codon G335 (Fwd primer, use w/oJT038 for C-term half) | CAGATAATGTATCAGGCACCANN SACATTATTTACGTTTCCGTCAC |
| oJT010 | Randomize rhesus TRIM5α at codon G335 (Rev primer, use w/oJT037 for N-term half) | GTGACGGAAACGTAAATAATGT SNNTGGTGCCTGATACATTATCTG |
| oJT011 | Randomize rhesus TRIM5α at codon T336 (Fwd primer, use w/oJT038 for C-term half) | ATGTATCAGGCACCAGGGNNS TTATTTACGTTTCCGTCACTCAC |
| oJT012 | Randomize rhesus TRIM5α at codon T336 (Rev primer, use w/oJT037 for N-term half) | GTGAGTGACGGAAACGTAAA TAASNNCCCTGGTGCCTGATACAT |
| oJT013 | Randomize rhesus TRIM5α at codon L337 (Fwd primer, use w/oJT038 for C-term half) | TATCAGGCACCAGGGACANNS TTTACGTTTCCGTCACTCAC |
| oJT014 | Randomize rhesus TRIM5α at codon L337 (Rev primer, use w/oJT037 for N-term half) | GTGAGTGACGGAAACGTAAASN NTGTCCCTGGTGCCTGATA |
| oJT015 | Randomize rhesus TRIM5α at codon F338 (Fwd primer, use w/oJT038 for C-term half) | CAGGCACCAGGGACATTANN SACGTTTCCGTCACTCACG |
| oJT016 | Randomize rhesus TRIM5α at codon F338 (Rev primer, use w/oJT037 for N-term half) | CGTGAGTGACGGAAACGTSNN TAATGTCCCTGGTGCCTG |
| oJT017 | Randomize rhesus TRIM5α at codon T339 (Fwd primer, use w/oJT038 for C-term half) | AGGCACCAGGGACATTATTT NNSTTTCCGTCACTCACGAATTTC |
| oJT018 | Randomize rhesus TRIM5α at codon T339 (Rev primer, use w/oJT037 for N-term half) | GAAATTCGTGAGTGACGGAA ASNNAAATAATGTCCCTGGTGCCT |
| oJT019 | Randomize rhesus TRIM5α at codon F340 (Fwd primer, use w/oJT038 for C-term half) | CACCAGGGACATTATTTACGN NSCCGTCACTCACGAATTTCAAT |
| oJT020 | Randomize rhesus TRIM5α at codon F340 (Rev primer, use w/oJT037 for N-term half) | ATTGAAATTCGTGAGTGACGGSNN CGTAAATAATGTCCCTGGTG |
| oJT021 | Randomize rhesus TRIM5α at codon P341 (Fwd primer, use w/oJT038 for C-term half) | ACCAGGGACATTATTTACGTTTNN STCACTCACGAATTTCAATTATTGTA |

*Table 1 continued on next page*

*Table 1 continued*

| Primer | Use | Sequence |
| --- | --- | --- |
| oJT022 | Randomize rhesus TRIM5α at codon P341 (Rev primer, use w/oJT037 for N-term half) | TACAATAATTGAAATTCGTGAGTG ASNNAAACGTAAATAATGTCCCTGGT |
| oJT023 | Randomize rhesus TRIM5α at codon S342 (Fwd primer, use w/oJT038 for C-term half) | GGGACATTATTTACGTTTCCGNN SCTCACGAATTTCAATTATTGTACT |
| oJT024 | Randomize rhesus TRIM5α at codon S342 (Rev primer, use w/oJT037 for N-term half) | AGTACAATAATTGAAATTCGTG AGSNNCGGAAACGTAAATAATGTCCC |
| oJT025 | Randomize rhesus TRIM5α at codon L343 (Fwd primer, use w/oJT038 for C-term half) | GACATTATTTACGTTTCCGTCANN SACGAATTTCAATTATTGTACTGGC |
| oJT026 | Randomize rhesus TRIM5α at codon L343 (Rev primer, use w/oJT037 for N-term half) | GCCAGTACAATAATTGAAATTCGT SNNTGACGGAAACGTAAATAATGTC |
| oJT027 | Randomize rhesus TRIM5α at codon T344 (Fwd primer, use w/oJT038 for C-term half) | CATTATTTACGTTTCCGTCACTCNN SAATTTCAATTATTGTACTGGCGTC |
| oJT028 | Randomize rhesus TRIM5α at codon T344 (Rev primer, use w/oJT037 for N-term half) | GACGCCAGTACAATAATTGAAATTS NNGAGTGACGGAAACGTAAATAATG |
| oAS024 | Flanking primer for human DMS libraries (Rev), amplifies human TRIM5α 3′ of BamHI site for Gibson cloning with pQCXIP-TRIM5α digested with NotI and BamHI | AGCACATACCCCCAGGATCCAAGCAG |
| oAS002 | Amplify N-term half of human TRIM5α (WT or R332P) before codon G330, to hybridize w/randomized G330 C-term half (Rev primer, use w/oJT037) | ATATATTATCTGTGGTTTCGGAGAGC |
| oAS001 | Randomize human TRIM5α (WT) at codon G330 (Fwd primer, use w/oAS024 for C-term half) | GCTCTCCGAAACCACAGATAATATAT **NNS**GCACGAGGGACAAGATACC |
| oJT142 | Randomize human TRIM5α (R332P) at codon G330 (Fwd primer, use w/oAS024 for C-term half) | GCTCTCCGAAACCACAGATAATATAT nnsGCACcAGGGACAAGATACCA |
| oAS004 | Amplify N-term half of human TRIM5α (WT or R332P) before codon A331, to hybridize w/randomized A331 C-term half (Rev primer, use w/oJT037) | CCCATATATTATCTGTGGTTTCGG |
| oAS003 | Randomize human TRIM5α (WT) at codon A331 (Fwd primer, use w/oAS024 for C-term half) | CCGAAACCACAGATAATATATGGG **NNS**CGAGGGACAAGATACCAGA |
| oJT143 | Randomize human TRIM5α (R332P) at codon A331 (Fwd primer, use w/oAS024 for C-term half) | CCGAAACCACAGATAATATATGGG nnsCcAGGGACAAGATACCAGAC |
| oAS006 | Amplify N-term half of human TRIM5α (WT) before codon R332 to hybridize w/randomized R332 C-term half (Rev primer, use w/oJT037) | TGCCCCATATATTATCTGTGGTTTC |
| oAS005 | Randomize human TRIM5α (WT) at codon R332 (Fwd primer, use w/oAS024 for C-term half) | GAAACCACAGATAATATATGGGGCA **NNS**GGGACAAGATACCAGACATTTG |
| oAS008 | Amplify N-term half of human TRIM5α (WT) before codon G333 to hybridize w/randomized G333 C-term half (Rev primer, use w/oJT037) | TCGTGCCCCATATATTATCTGTG |
| oAS007 | Randomize human TRIM5α (WT) at codon G333 (Fwd primer, use w/oAS024 for C-term half) | CACAGATAATATATGGGGCACGA**NN S**ACAAGATACCAGACATTTGTGAATT |
| oJT144 | Amplify N-term half of human TRIM5α (R332P) before codon G333 to hybridize w/randomized G333 C-term half (Rev primer, use w/oJT037) | TgGTGCCCCATATATTATCTGTG |

*Table 1 continued on next page*

*Table 1 continued*

| Primer | Use | Sequence |
|---|---|---|
| oJT145 | Randomize human TRIM5α (R332P) at codon G333 (Fwd primer, use w/oAS024 for C-term half) | CACAGATAATATATGGGGCACcAnns ACAAGATACCAGACATTTGTGAATTTC |
| oAS010 | Amplify N-term half of human TRIM5α (WT) before codon T334 to hybridize w/randomized T334 C-term half (Rev primer, use w/oJT037) | CCCTCGTGCCCCATATATTA |
| oAS009 | Randomize human TRIM5α (WT) at codon T334 (Fwd primer, use w/oAS024 for C-term half) | TAATATATGGGGCACGAGGGNN SAGATACCAGACATTTGTGAATTTCA |
| oJT146 | Amplify N-term half of human TRIM5α (R332P) before codon T334 to hybridize w/randomized T334 C-term half (Rev primer, use w/oJT037) | CCCTgGTGCCCCATATATTATCTG |
| oJT147 | Randomize human TRIM5α (R332P) at codon T334 (Fwd primer, use w/oAS024 for C-term half) | CAGATAATATATGGGGCACcAGGGnns AGATACCAGACATTTGTGAATTTCAATTA |
| oAS012 | Amplify N-term half of human TRIM5α (WT) before codon R335 to hybridize w/randomized R335 C-term half (Rev primer, use w/oJT037) | TGTCCCTCGTGCCCCAT |
| oAS011 | Randomize human TRIM5α (WT) at codon R335 (Fwd primer, use w/oAS024 for C-term half) | ATggggcacgagggaca NNStaccagacatttgtgAATTTCAATTATTG |
| oJT148 | Amplify N-term half of human TRIM5α (R332P) before codon R335 to hybridize w/randomized R335 C-term half (Rev primer, use w/oJT037) | TGTCCCTgGTGCCCCATATA |
| oJT149 | Randomize human TRIM5α (R332P) at codon R335 (Fwd primer, use w/oAS024 for C-term half) | TATATGGGGCACcAGGGACAnns TACCAGACATTTGTGAATTTCAATTATTG |
| oAS014 | Amplify N-term half of human TRIM5α (WT) before codon Y336 to hybridize w/randomized Y336 C-term half (Rev primer, use w/oJT037) | TCTTGTCCCTCGTGCCC |
| oAS013 | Randomize human TRIM5α (WT) at codon Y336 (Fwd primer, use w/oAS024 for C-term half) | GGGCACGAGGGACAAGANNS CAGACATTTGTGAATTTCAATTATTG |
| oJT150 | Amplify N-term half of human TRIM5α (R332P) before codon Y336 to hybridize w/randomized Y336 C-term half (Rev primer, use w/oJT037) | TCTTGTCCCTgGTGCCCC |
| oJT151 | Randomize human TRIM5α (R332P) at codon Y336 (Fwd primer, use w/oAS024 for C-term half) | GGGGCACcAGGGACAAGAnnsCA GACATTTGTGAATTTCAATTATTGTAC |
| oAS016 | Amplify N-term half of human TRIM5α (WT) before codon Q337 to hybridize w/randomized Q337 C-term half (Rev primer, use w/oJT037) | GTATCTTGTCCCTCGTGCC |
| oAS015 | Randomize human TRIM5α (WT) at codon Q337 (Fwd primer, use w/oAS024 for C-term half) | GCACGAGGGACAAGATACNNS ACATTTGTGAATTTCAATTATTGTACTG |
| oJT152 | Amplify N-term half of human TRIM5α (R332P) before codon Q337 to hybridize w/randomized Q337 C-term half (Rev primer, use w/oJT037) | GTATCTTGTCCCTgGTGCC |
| oJT153 | Randomize human TRIM5α (R332P) at codon Q337 (Fwd primer, use w/oAS024 for C-term half) | GGCACcAGGGACAAGATACnns ACATTTGTGAATTTCAATTATTGTACTGG |

*Table 1 continued*

| Primer | Use | Sequence |
|--------|-----|----------|
| oAS018 | Amplify N-term half of human TRIM5α (WT) before codon T338 to hybridize w/randomized T338 C-term half (Rev primer, use w/oJT037) | CTGGTATCTTGTCCCTCGTG |
| oAS017 | Randomize human TRIM5α (WT) at codon T338 (Fwd primer, use w/oAS024 for C-term half) | CACGAGGGACAAGATACCAG**NNS**TTTGTGAATTTCAATTATTGTACTGG |
| oJT154 | Amplify N-term half of human TRIM5α (R332P) before codon T338 to hybridize w/randomized T338 C-term half (Rev primer, use w/oJT037) | CTGGTATCTTGTCCCTgGTG |
| oJT155 | Randomize human TRIM5α (R332P) at codon T338 (Fwd primer, use w/oAS024 for C-term half) | CACcAGGGACAAGATACCAGnnsTTTGTGAATTTCAATTATTGTACTGGCAT |
| oAS020 | Amplify N-term half of human TRIM5α (WT) before codon F339 to hybridize w/randomized F339 C-term half (Rev primer, use w/oJT037) | TGTCTGGTATCTTGTCCCTC |
| oAS019 | Randomize human TRIM5α (WT) at codon F339 (Fwd primer, use w/oAS024 for C-term half) | GAGGGACAAGATACCAGACA**NNS**GTGAATTTCAATTATTGTACTGGC |
| oJT156 | Amplify N-term half of human TRIM5α (R332P) before codon F339 to hybridize w/randomized F339 C-term half (Rev primer, use w/oJT037) | TGTCTGGTATCTTGTCCCTgG |
| oJT157 | Randomize human TRIM5α (R332P) at codon F339 (Fwd primer, use w/oAS024 for C-term half) | CcAGGGACAAGATACCAGACAnnsGTGAATTTCAATTATTGTACTGGCATC |
| oAS022 | Amplify N-term half of human TRIM5α (WT) before codon V340 to hybridize w/randomized V340 C-term half (Rev primer, use w/oJT037) | AAATGTCTGGTATCTTGTCCCTC |
| oAS021 | Randomize human TRIM5α (WT) at codon V340 (Fwd primer, use w/oAS024 for C-term half) | AGGGACAAGATACCAGACATTT**NNS**AATTTCAATTATTGTACTGGCATCC |
| oJT158 | Amplify N-term half of human TRIM5α (R332P) before codon V340 to hybridize w/randomized V340 C-term half (Rev primer, use w/oJT037) | AAATGTCTGGTATCTTGTCCCTg |
| oJT159 | Randomize human TRIM5α (R332P) at codon V340 (Fwd primer, use w/oAS024 for C-term half) | cAGGGACAAGATACCAGACATTTnnsAATTTCAATTATTGTACTGGCATCCTG |
| Illumina library construction | | |
| oJT055 | Illumina library construction, PCR1 Fwd primer (rhesus TRIM5α only), amplifies v1 loop and adds adaptor | tcgtcggcagcgtcagatgtgtataagagacagTGAGCTCTCGGAACCCACAGATAATGTAT |
| oJT056 | Illumina library construction, PCR1 Rev primer (rhesus TRIM5α only), amplifies v1 loop and adds adaptor | gtctcgtgggctcggagatgtgtataagagacagGCCCAGGACGCCAGTACAATAATTGAAATT |
| oJT113 | Illumina library construction, PCR1 Fwd primer (human TRIM5α libraries) amplifies v1 loop and adds adaptor | tcgtcggcagcgtcagatgtgtataagagacagCAAGTGAGCTCTCCGAAACCACAGATAATATAT |
| oJT114 | Illumina library construction, PCR1 Rev primer (human TRIM5α libraries), amplifies v1 loop and adds adaptor | gtctcgtgggctcggagatgtgtataagagacagGAGCCCAGGATGCCAGTACAATAATTGAAATT |
| oJT057 | Illumina library construction, PCR2 Fwd primer (all libraries), adds P5 adaptor | AATGATACGGCGACCACCGAGATCTACACtagatcgcTCGTCGGCAGCGTC |
| oJT058 | Illumina library construction, PCR2 Rev primer (all libraries), adds P7 adaptor and N701 barcoode | CAAGCAGAAGACGGCATACGAGATtcgccttaGTCTCGTGGGCTCGG |

*Table 1 continued on next page*

*Table 1 continued*

| Primer | Use | Sequence |
| --- | --- | --- |
| oJT059 | Illumina library construction, PCR2 Rev primer (all libraries), adds P7 adaptor and N702 barcoode | CAAGCAGAAGACGGCATACGAGAT ctagtacgGTCTCGTGGGCTCGG |
| oJT060 | Illumina library construction, PCR2 Rev primer (all libraries), adds P7 adaptor and N703 barcoode | CAAGCAGAAGACGGCATACGAGAT ttctgcctGTCTCGTGGGCTCGG |
| oJT115 | Illumina library construction, PCR2 Rev primer (all libraries), adds P7 adaptor and N704 barcoode | CAAGCAGAAGACGGCATACGAGA TgctcaggaGTCTCGTGGGCTCGG |
| oJT116 | Illumina library construction, PCR2 Rev primer (all libraries), adds P7 adaptor and N705 barcoode | CAAGCAGAAGACGGCATACGAGA TaggagtccGTCTCGTGGGCTCGG |
| oJT117 | Illumina library construction, PCR2 Rev primer (all libraries), adds P7 adaptor and N706 barcoode | CAAGCAGAAGACGGCATACGAGA TcatgcctaGTCTCGTGGGCTCGG |
| oJT118 | Illumina library construction, PCR2 Rev primer (all libraries), adds P7 adaptor and N707 barcoode | CAAGCAGAAGACGGCATACGAGA TgtagagagGTCTCGTGGGCTCGG |
| oJT119 | Illumina library construction, PCR2 Rev primer (all libraries), adds P7 adaptor and N708 barcoode | CAAGCAGAAGACGGCATACGAGA TcctctctgGTCTCGTGGGCTCGG |
| oJT138 | Illumina library construction, PCR2 Rev primer (all libraries), adds P7 adaptor and N709 barcoode | CAAGCAGAAGACGGCATACGAGA TagcgtagcGTCTCGTGGGCTCGG |
| oJT139 | Illumina library construction, PCR2 Rev primer (all libraries), adds P7 adaptor and N710 barcoode | CAAGCAGAAGACGGCATACGAGA TcagcctcgGTCTCGTGGGCTCGG |
| oJT140 | Illumina library construction, PCR2 Rev primer (all libraries), adds P7 adaptor and N711 barcoode | CAAGCAGAAGACGGCATACGAGA TtgcctcttGTCTCGTGGGCTCGG |
| oJT141 | Illumina library construction, PCR2 Rev primer (all libraries), adds P7 adaptor and N712 barcoode | CAAGCAGAAGACGGCATACGAGA TtcctctacGTCTCGTGGGCTCGG |
| oJT166 | Illumina library construction, PCR2 Rev primer (all libraries), adds P7 adaptor and N714 barcoode | CAAGCAGAAGACGGCATACGAGA TTCATGAGCGTCTCGTGGGCTCGG |
| oJT167 | Illumina library construction, PCR2 Rev primer (all libraries), adds P7 adaptor and N715 barcoode | CAAGCAGAAGACGGCATACGAGA TCCTGAGATGTCTCGTGGGCTCGG |
| oJT168 | Illumina library construction, PCR2 Rev primer (all libraries), adds P7 adaptor and N716 barcoode | CAAGCAGAAGACGGCATACGAGA TTAGCGAGTGTCTCGTGGGCTCGG |
| oJT169 | Illumina library construction, PCR2 Rev primer (all libraries), adds P7 adaptor and N718 barcoode | CAAGCAGAAGACGGCATACGAGA TGTAGCTCCGTCTCGTGGGCTCGG |
| oJT170 | Illumina library construction, PCR2 Rev primer (all libraries), adds P7 adaptor and N719 barcoode | CAAGCAGAAGACGGCATACGAGA TTACTACGCGTCTCGTGGGCTCGG |
| oJT171 | Illumina library construction, PCR2 Rev primer (all libraries), adds P7 adaptor and N720 barcoode | CAAGCAGAAGACGGCATACGAGA TAGGCTCCGGTCTCGTGGGCTCGG |
| oJT172 | Illumina library construction, PCR2 Rev primer (all libraries), adds P7 adaptor and N721 barcoode | AAGCAGAAGACGGCATACGAGA TGCAGCGTAGTCTCGTGGGCTCGG |
| oJT173 | Illumina library construction, PCR2 Rev primer (all libraries), adds P7 adaptor and N722 barcoode | CAAGCAGAAGACGGCATACGAGAT CTGCGCATGTCTCGTGGGCTCGG |

*Table 1 continued*

| Primer | Use | Sequence |
|---|---|---|
| oJT174 | Illumina library construction, PCR2 Rev primer (all libraries), adds P7 adaptor and N723 barcoode | CAAGCAGAAGACGGCATACGAGAT GAGCGCTAGTCTCGTGGGCTCGG |
| oJT175 | Illumina library construction, PCR2 Rev primer (all libraries), adds P7 adaptor and N724 barcoode | CAAGCAGAAGACGGCATACGAGAT CGCTCAGTGTCTCGTGGGCTCGG |
| oJT176 | Illumina library construction, PCR2 Rev primer (all libraries), adds P7 adaptor and N726 barcoode | CAAGCAGAAGACGGCATACGAGAT GTCTTAGGGTCTCGTGGGCTCGG |
| oJT177 | Illumina library construction, PCR2 Rev primer (all libraries), adds P7 adaptor and N727 barcoode | CAAGCAGAAGACGGCATACGAGAT ACTGATCGGTCTCGTGGGCTCGG |
| oJT178 | Illumina library construction, PCR2 Rev primer (all libraries), adds P7 adaptor and N728 barcoode | CAAGCAGAAGACGGCATACGAGAT TAGCTGCAGTCTCGTGGGCTCGG |
| oJT179 | Illumina library construction, PCR2 Rev primer (all libraries), adds P7 adaptor and N729 barcoode | CAAGCAGAAGACGGCATACGAGAT GACGTCGAGTCTCGTGGGCTCGG |
| RhT5-IlluminaF | Custom sequencing primer for rhesus TRIM5α Illumina libraries, sequences rhesus TRIM5α v1 loop (39 nt) | TGAGCTCTCGGAACCC ACAGATAATGTAT |
| HsT5-IlluminaF | Custom sequencing primer for human TRIM5α Illumina libraries, sequences human TRIM5α v1 loop (33 nt) | CAAGTGAGCTCTCCGAAA CCACAGATAATATAT |
| Quikchange PCR for targeted mutagenesis | | |
| oCY001 | Generate G330E mutation in human TRIM5α (Fwd primer, amplify full plasmid with oCY002) | CTCCGAAACCACAGATAATATATG aGGCACGAGGGACAAGATAC |
| oCY002 | Generate G330E mutation in human TRIM5α (Rev primer, amplify full plasmid with oCY001) | GTATCTTGTCCCTCGTGCCt CATATATTATCTGTGGTTTCGGAG |
| oCY003 | Generate A331E mutation in human TRIM5α F(wd primer, amplify full plasmid with oCY004) | GAAACCACAGATAATATAT GGGGaACGAGGGACAAGATACCAG |
| oCY004 | Generate A331E mutation in human TRIM5α (Rev primer, amplify full plasmid with oCY003) | CTGGTATCTTGTCCCTCGTt CCCCATATATTATCTGTGGTTTC |
| oCY005 | Generate R332E mutation in human TRIM5α (Fwd primer, amplify full plasmid with oCY006) | ACCACAGATAATATATGGGGCAga AGGGACAAGATACCAGACATT |
| oCY006 | Generate R332E mutation in human TRIM5α (Rev primer, amplify full plasmid with oCY005) | AATGTCTGGTATCTTGTCCCTtc TGCCCCATATATTATCTGTGGT |
| oJT040 | Generate R332P mutation in human TRIM5α (Fwd primer, amplify full plasmid with oJT041) | CCACAGATAATATATGGGGCAC cAGGGACAAGATACCAGACATTTG |
| oJT041 | Generate R332P mutation in human TRIM5α (Rev primer, amplify full plasmid with oJT040) | CAAATGTCTGGTATCTTGTCCCT gGTGCCCCATATATTATCTGTGG |
| oCY007 | Generate G333Y mutation in human TRIM5α (Fwd primer, amplify full plasmid with oCY008) | CACAGATAATATATGGGGCACGA tacACAAGATACCAGACATTTGTGAATT |
| oCY008 | Generate G333Y mutation in human TRIM5α (Rev primer, amplify full plasmid with oCY007) | AATTCACAAATGTCTGGTATCTTGTgta TCGTGCCCCATATATTATCTGTG |
| oCY009 | Generate G333D mutation in human TRIM5α (Fwd primer, amplify full plasmid with oCY010) | CAGATAATATATGGGGCACGAGat ACAAGATACCAGACATTTGTGAATTTC |
| oCY010 | Generate G333D mutation in human TRIM5α (Rev primer, amplify full plasmid with oCY009) | GAAATTCACAAATGTCTGGTATCT TGTatCTCGTGCCCCATATATTATCTG |
| oCY011 | Generate T334D mutation in human TRIM5α (Fwd primer, amplify full plasmid with oCY012) | GATAATATATGGGGCACGAGGG gacAGATACCAGACATTTGTGAATTTC |
| oCY012 | Generate T334D mutation in human TRIM5α (Rev primer, amplify full plasmid with oCY011) | GAAATTCACAAATGTCTGGTATC TgtcCCCTCGTGCCCCATATATTATC |
| oJT246 | Generate R335A mutation in human TRIM5α (Fwd primer, amplify full plasmid with oJT247) | TATGGGGCACGAGGGACAgcATA CCAGACATTTGTGAATTTCAATTATTG |

*Table 1 continued on next page*

*Table 1 continued*

| Primer | Use | Sequence |
| --- | --- | --- |
| oJT247 | Generate R335A mutation in human TRIM5α (Rev primer, amplify full plasmid with oJT246) | CAATAATTGAAATTCACAAATGTCT GGTATgcTGTCCCTCGTGCCCCATA |
| oCY013 | Generate R335E mutation in human TRIM5α (Fwd primer, amplify full plasmid with oCY014) | TGGGGCACGAGGGACAgaATACCA GACATTTGTGAATTTCAATTATTG |
| oCY014 | Generate R335E mutation in human TRIM5α (Rev primer, amplify full plasmid with oCY013) | CAATAATTGAAATTCACAAATGT CTGGTATtcTGTCCCTCGTGCCCCA |
| oCY015 | Generate Y336E mutation in human TRIM5α (Fwd primer, amplify full plasmid with oCY016) | GGGCACGAGGGACAAGAgAaCA GACATTTGTGAATTTCAATTATTGTAC |
| oCY016 | Generate Y336E mutation in human TRIM5α (Rev primer, amplify full plasmid with oCY015) | GTACAATAATTGAAATTCACAAAT GTCTGtTcTCTTGTCCCTCGTGCCC |
| oCY017 | Generate Q337D mutation in human TRIM5α (Fwd primer, amplify full plasmid with oCY018) | GGCACGAGGGACAAGATACgAtAC ATTTGTGAATTTCAATTATTGTACTG |
| oCY018 | Generate Q337D mutation in human TRIM5α (Rev primer, amplify full plasmid with oCY017) | CAGTACAATAATTGAAATTCACAAA TGTaTcGTATCTTGTCCCTCGTGCC |
| oAS2019-05 | Generate Q337N mutation in human TRIM5α (Fwd primer, amplify full plasmid with oAS2019-06) | GGCACGAGGGACAAGATACaacACAT TTGTGAATTTCAATTATTGTACTGG |
| oAS2019-06 | Generate Q337N mutation in human TRIM5α (Rev primer, amplify full plasmid with oAS2019-05) | CCAGTACAATAATTGAAATTCACAAA TGTgttGTATCTTGTCCCTCGTGCC |
| oCY019 | Generate T338E mutation in human TRIM5α (Fwd primer, amplify full plasmid with oCY020) | ACGAGGGACAAGATACCAGgaATTT GTGAATTTCAATTATTGTACTGGC |
| oCY020 | Generate T338E mutation in human TRIM5α (Rev primer, amplify full plasmid with oCY019) | GCCAGTACAATAATTGAAATTCAC AAATtcCTGGTATCTTGTCCCTCGT |
| oCY021 | Generate F339E mutation in human TRIM5α (Fwd primer, amplify full plasmid with oCY022) | GAGGGACAAGATACCAGACAgaa GTGAATTTCAATTATTGTACTGGCA |
| oCY022 | Generate F339E mutation in human TRIM5α (Rev primer, amplify full plasmid with oCY021) | TGCCAGTACAATAATTGAAATTCAC ttcTGTCTGGTATCTTGTCCCTC |
| oCY023 | Generate V340E mutation in human TRIM5α (Fwd primer, amplify full plasmid with oCY024) | GGGACAAGATACCAGACATTT GaGAATTTCAATTATTGTACTGGCATCC |
| oCY024 | Generate V340E mutation in human TRIM5α (Rev primer, amplify full plasmid with oCY023) | GGATGCCAGTACAATAATTGAAAT TCtCAAATGTCTGGTATCTTGTCCC |
| oAS2019-07 | Generate V340H mutation in human TRIM5α (Fwd primer, amplify full plasmid with oAS2019-08) | GAGGGACAAGATACCAGACATTT cacAATTTCAATTATTGTACTGGCATCCT |
| oAS2019-08 | Generate V340H mutation in human TRIM5α (Rev primer, amplify full plasmid with oAS2019-07) | AGGATGCCAGTACAATAATTGAAA TTgtgAAATGTCTGGTATCTTGTCCCTC |
| oJT162 | Generate V340stop mutation in human TRIM5α (Fwd primer, amplify full plasmid with oJT163) | GAGGGACAAGATACCAGACATTT taGAATTTCAATTATTGTACTGGCATCC |
| oJT163 | Generate V340stop mutation in human TRIM5α (Rev primer, amplify full plasmid with oJT162) | GGATGCCAGTACAATAATTGAAATT CtaAAATGTCTGGTATCTTGTCCCTC |
| oJT248 | Generate R335A double mutation in human TRIM5α-R332P (Fwd primer, amplify full plasmid with oJT249) | TATGGGGCACcAGGGACAgc ATACCAGACATTTGTGAATTTCAATTATTG |
| oJT249 | Generate R335A double mutation in human TRIM5α-R332P (Rev primer, amplify full plasmid with oJT248) | CAATAATTGAAATTCACAAATGTCT GGTATgcTGTCCCTgGTGCCCCATA |

*Table 1 continued on next page*

*Table 1 continued*

| Primer | Use | Sequence |
|---|---|---|
| oAS2019-03 | Generate R335E double mutation in human TRIM5α-R332P (Fwd primer, amplify full plasmid with oAS2019-04) | TGGGGCACcAGGGACAgaaTACC AGACATTTGTGAATTTCAATTATTG |
| oAS2019-04 | Generate R335E double mutation in human TRIM5α-R332P (Rev primer, amplify full plasmid with oAS2019-03) | CAATAATTGAAATTCACAAATGTC TGGTAttcTGTCCCTgGTGCCCCA |
| oAS2019-01 | Generate Q337D double mutation in human TRIM5α-R332P (Fwd primer, amplify full plasmid with oAS2019-02) | GGCACcAGGGACAAGATACgacAC ATTTGTGAATTTCAATTATTGTACTGG |
| oAS2019-02 | Generate Q337D double mutation in human TRIM5α-R332P (Rev primer, amplify full plasmid with oAS2019-01) | CCAGTACAATAATTGAAATTCACAA ATGTgtcGTATCTTGTCCCTgGTGCC |
| oAS2019-09 | Generate Q337N double mutation in human TRIM5α-R332P (Fwd primer, amplify full plasmid with oAS2019-10) | GGCACcAGGGACAAGATACaac ACATTTGTGAATTTCAATTATTGTACTGG |
| oAS2019-10 | Generate Q337N double mutation in human TRIM5α-R332P (Rev primer, amplify full plasmid with oAS2019-09) | CCAGTACAATAATTGAAATTCA CAAATGTgttGTATCTTGTCCCTgGTGCC |
| oJT122 | Generate Q332E mutation in rhesus TRIM5α (Fwd primer, amplify full plasmid with oJT123) | CTCGGAACCCACAGATAATGTATg AGGCACCAGGGACATTATTTAC |
| oJT123 | Generate Q332E mutation in rhesus TRIM5α (Rev primer, amplify full plasmid with oJT122) | GTAAATAATGTCCCTGGTGCCT cATACATTATCTGTGGGTTCCGAG |
| oJT120 | Generate A333E mutation in rhesus TRIM5α (Fwd primer, amplify full plasmid with oJT121) | GAACCCACAGATAATGTATCAGG aACCAGGGACATTATTTACGTTTCC |
| oJT121 | Generate A333E mutation in rhesus TRIM5α (Rev primer, amplify full plasmid with oJT120) | GGAAACGTAAATAATGTCCCTGGT tCCTGATACATTATCTGTGGGTTC |
| oCY063 | Generate A333W mutation in rhesus TRIM5α (Fwd primer, amplify full plasmid with oCY064) | GAACCCACAGATAATGTATCAGtgg CCAGGGACATTATTTACGTTTC |
| oCY064 | Generate A333W mutation in rhesus TRIM5α (Rev primer, amplify full plasmid with oCY063) | GAAACGTAAATAATGTCCCTGGCC ACTGATACATTATCTGTGGGTTC |
| oJT124 | Generate P334M mutation in rhesus TRIM5α (Fwd primer, amplify full plasmid with oJT125) | CCCACAGATAATGTATCAGGCAatg GGGACATTATTTACGTTTCCGTC |
| oJT125 | Generate P334M mutation in rhesus TRIM5α (Rev primer, amplify full plasmid with oJT124) | GACGGAAACGTAAATAATGTCCCcat TGCCTGATACATTATCTGTGGG |
| oJT095 | Generate G335I mutation in rhesus TRIM5α (Fwd primer, amplify full plasmid with oJT096) | ACAGATAATGTATCAGGCACCAatc ACATTATTTACGTTTCCGTCACTC |
| oJT096 | Generate G335I mutation in rhesus TRIM5α (Rev primer, amplify full plasmid with oJT095) | GAGTGACGGAAACGTAAATAATGTgat TGGTGCCTGATACATTATCTGT |
| oJT097 | Generate T336Q mutation in rhesus TRIM5α (Fwd primer, amplify full plasmid with oJT098) | GATAATGTATCAGGCACCAGGGcaa TTATTTACGTTTCCGTCACTCAC |
| oJT098 | Generate T336Q mutation in rhesus TRIM5α (Rev primer, amplify full plasmid with oJT097) | GTGAGTGACGGAAACGTAAATAA ttgCCCTGGTGCCTGATACATTATC |
| oJT099 | Generate L337N mutation in rhesus TRIM5α (Fwd primer, amplify full plasmid with oJT100) | GTATCAGGCACCAGGGACAaac TTTACGTTTCCGTCACTCACG |

*Table 1 continued on next page*

Table 1 continued

| Primer | Use | Sequence |
| --- | --- | --- |
| oJT100 | Generate L337N mutation in rhesus TRIM5α (Rev primer, amplify full plasmid with oJT99) | CGTGAGTGACGGAAACGTAAAgtt TGTCCCTGGTGCCTGATAC |
| oJT103 | Generate F338K mutation in rhesus TRIM5α (Fwd primer, amplify full plasmid with oJT104) | TCAGGCACCAGGGACATTAaag ACGTTTCCGTCACTCACG |
| oJT104 | Generate F338K mutation in rhesus TRIM5α (Rev primer, amplify full plasmid with oJT103) | CGTGAGTGACGGAAACGTctt TAATGTCCCTGGTGCCTGA |
| oJT126 | Generate F338Q mutation in rhesus TRIM5α (Fwd primer, amplify full plasmid with oJT127) | TCAGGCACCAGGGACATTAcag ACGTTTCCGTCACTCACGA |
| oJT127 | Generate F338Q mutation in rhesus TRIM5α (Rev primer, amplify full plasmid with oJT126) | TCGTGAGTGACGGAAACGTctg TAATGTCCCTGGTGCCTGA |
| oJT105 | Generate T339F mutation in rhesus TRIM5α (Fwd primer, amplify full plasmid with oJT106) | CAGGCACCAGGGACATTATTT ttcTTTCCGTCACTCACGAATTTCA |
| oJT106 | Generate T339F mutation in rhesus TRIM5α (Rev primer, amplify full plasmid with oJT105) | TGAAATTCGTGAGTGACGGAAA gaaAAATAATGTCCCTGGTGCCTG |
| oJT128 | Generate T339Q mutation in rhesus TRIM5α (Fwd primer, amplify full plasmid with oJT129) | CAGGCACCAGGGACATTATTT caGTTTCCGTCACTCACGAATTTC |
| oJT129 | Generate T339Q mutation in rhesus TRIM5α (Rev primer, amplify full plasmid with oJT128) | GAAATTCGTGAGTGACGGAAACtg AAATAATGTCCCTGGTGCCTG |
| oJT130 | Generate F340D mutation in rhesus TRIM5α (Fwd primer, amplify full plasmid with oJT131) | GCACCAGGGACATTATTTACGga TCCGTCACTCACGAATTTCAATTA |
| oJT131 | Generate F340D mutation in rhesus TRIM5α (Rev primer, amplify full plasmid with oJT130) | TAATTGAAATTCGTGAGTGACGG AtcCGTAAATAATGTCCCTGGTGC |
| oJT132 | Generate P341G mutation in rhesus TRIM5α (Fwd primer, amplify full plasmid with oJT133) | CACCAGGGACATTATTTACGTTT ggGTCACTCACGAATTTCAATTATTGT |
| oJT133 | Generate P341G mutation in rhesus TRIM5α (Rev primer, amplify full plasmid with oJT132) | ACAATAATTGAAATTCGTGAGTGAC ccAAACGTAAATAATGTCCCTGGTG |
| oJT107 | Generate P341I mutation in rhesus TRIM5α (Fwd primer, amplify full plasmid with oJT108) | CACCAGGGACATTATTTACGTT TataTCACTCACGAATTTCAATTATTGTAC |
| oJT108 | Generate P341I mutation in rhesus TRIM5α (Rev primer, amplify full plasmid with oJT107) | GTACAATAATTGAAATTCGTGAGT GAtatAAACGTAAATAATGTCCCTGGTG |
| oJT109 | Generate P341IK mutation in rhesus TRIM5α (Fwd primer, amplify full plasmid with oJT110) | CACCAGGGACATTATTTACGTTT aaaTCACTCACGAATTTCAATTATTGTAC |
| oJT110 | Generate P341IK mutation in rhesus TRIM5α (Rev primer, amplify full plasmid with oJT109) | GTACAATAATTGAAATTCGTGAGTG AtttAAACGTAAATAATGTCCCTGGTG |
| oJT134 | Generate S342G mutation in rhesus TRIM5α (Fwd primer, amplify full plasmid with oJT135) | AGGGACATTATTTACGTTTCCGgg ACTCACGAATTTCAATTATTGTACTG |
| oJT135 | Generate S342G mutation in rhesus TRIM5α (Rev primer, amplify full plasmid with oJT134) | CAGTACAATAATTGAAATTCGTG AGTccCGGAAACGTAAATAATGTCCCT |

Table 1 continued on next page

*Table 1 continued*

| Primer | Use | Sequence |
| --- | --- | --- |
| oJT101 | Generate T344E mutation in rhesus TRIM5α (Fwd primer, amplify full plasmid with oJT102) | CATTATTTACGTTTCCGTCACTC gagAATTTCAATTATTGTACTGGCGTC |
| oJT102 | Generate T344E mutation in rhesus TRIM5α (Rev primer, amplify full plasmid with oJT101) | GACGCCAGTACAATAATTGAAATT ctcGAGTGACGGAAACGTAAATAATG |
| oJT111 | Generate T344stop mutation in rhesus TRIM5α (Fwd primer, amplify full plasmid with oJT112) | CATTATTTACGTTTCCGTCACTC tagAATTTCAATTATTGTACTGGCGTC |
| oJT112 | Generate T344stop mutation in rhesus TRIM5α (Rev primer, amplify full plasmid with oJT111) | GACGCCAGTACAATAATTGAAATT ctaGAGTGACGGAAACGTAAATAATG |
| Sequencing primers | | |
| pQCXIP-F | Sequencing primer from pQCXIP backbone 5' of multiple cloning site (Fwd) | acaccgggaccgatccag |
| HsT5-midF | Sequencing primer from midpoint of human TRIM5α (Fwd) | GATCTGGAGCATCGGCTG |
| HsT5-midR | Sequencing primer from midpoint of human TRIM5α (Rev) | CAAGGTCACGTTCTCCGTC |
| RhT5-midF | Sequencing primer from midpoint of rhesus TRIM5α (Fwd) | CTCATCTCAGAACTGGAGCATC |
| RhT5-midR | Sequencing primer from midpoint of rhesus TRIM5α (Rev) | CTTCAAGGTCATGTTCTCAATCC |

into a single fragment using the same flanking primers as in the first amplification. PCR products were gel purified and cloned via Gibson assembly (NEB) into pQCXIP-TRIM5α-HA of the matching species, which had been digested with NotI and BamHI and gel purified. Gibson assembly products were transformed into high-efficiency chemically competent DH5α (NEB) with 30 min of heat shock recovery. Serial dilutions were plated to count the number of unique colonies, and transformations were repeated until at least 100x library coverage was achieved (human: $32 \times 11 \times 100 = 3.52 \times 10^4$ colonies; rhesus: $32 \times 13 \times 100 = 4.16 \times 10^4$ colonies). To ensure library quality, 40 random colonies were sequenced from each library. Clones were verified to have insert by analytical restriction digest, and the coding sequence was fully sequenced to ensure that (1) each clone had only one mutation, (2) there were no mutations outside the v1 loop, and (3) the number of sites mutated once, twice, etc. among these 40 clones approximated a Poisson distribution. When libraries met these criteria, colonies were scraped from all transformation plates and plasmids were directly purified, without further growth to avoid amplification bias, using NucleoBond Xtra midiprep kits (Takara Bio).

## Cell lines

HEK-293T/17 (RRID:CVCL_1926) and CRFK (RRID:CVCL_2426) cells were purchased from ATCC (Manassas, VA) and not further authenticated; cells were confirmed to be mycoplasma free by MycoProbe kit (R and D Systems, Minneapolis, MN). Cells were grown on tissue-culture treated plates in high-glucose and L-glutamine containing DMEM (Thermo Fisher, Waltham, MA) supplemented with 1x penicillin/streptomycin (Thermo Fisher) and 10% fetal bovine serum (Thermo Fisher). Cells were grown at 37°C, 5% $CO_2$ in humidified incubators and passaged by digestion with 0.05% trypsin-EDTA (Thermo Fisher). Cell counting was performed using a TC20 automated cell counter (BioRad, Hercules, CA).

## Virus production, titering, and transduction

HEK-293T/17 were seeded at $5 \times 10^5$ cells/well in 6-well plates the day prior to transfection. Transfections were performed with Trans-IT 293T transfection reagent (Mirus Bio, Madison, WI) according to manufacturer's instructions, using 3 μL reagent per μg DNA. All transfected DNA was purified using PureYield mini or NuceloBond midi kits to minimize LPS contamination and quantified by NanoDrop (Thermo Fisher) A260. To produce HIV-1, each well was transfected with 1 μg of p8.9NdSB, 667 ng of pHIV-ZsGreen, and 333 ng of pMD2.G. N-MLV transfections contained 1 μg of pQCXIP-eGFP, 667 ng of pCIG3N, and 333 ng of pMD2.G. HIV-2, SIVcpz, and SIVmac transfections contained 1 μg of pALPS-eGFP, 333 ng of pMD2.G, and 667 ng of either p8.9NdSB HIV-2 CA, pHIV-Gb2, or pHIV-MAC, respectively. TRIM5α-transducing virus was produced using 1 μg of the appropriate TRIM5α construct, 600 ng of JK3, 300 ng of L-VSV-G, and 100 ng of CMV-Tat. After 24 hr, media was replaced with 1 mL of fresh media. Virus was harvested at 48 hr post-transfection. To harvest, cells were pelleted from virus-containing media at 500 x g, and supernatant was removed, aliquoted, and snap frozen in liquid nitrogen. To increase titers for HIV-2 and N-MLV, and in some cases HIV-1, virus was concentrated prior to freezing. To concentrate, virus was pelleted through a 20% sucrose cushion at 23,000 rpm (~70,000 x g) for 1 hr at 4°C. Pellets were air dried for 5 min, and then resuspended in fresh media for 24 hr with periodic gentle vortexing.

All viruses were titered under conditions most closely mimicking their large-scale use. CRFK cells were seeded at $1 \times 10^5$ cells/mL the day prior to transduction. Freshly thawed viruses were serially diluted and replaced cellular media at ½ x volume. No transducing reagent was used for GFP-marked retroviral VLPs; TRIM5α-transducing VLPs were supplemented with 10 μg/mL polybrene. Plates were centrifuged at 1100 x g for 30 min and then incubated at 37°C. The following day, virus was removed and cells were fed fresh media, which contained 6 μg/mL puromycin for TRIM5α-transducing VLPs only. For GFP-marked retroviral VLPs, transduction efficiency was monitored by flow cytometry 72 hr after transduction. For TRIM5α-transducing VLPs, cell survival was monitored daily by estimating cell confluence, until untransduced cells were completely dead (no surface-adhered cells). Media was replaced with fresh puromycin-containing media every 2–3 days, and cells were passaged into larger well format as needed. Multiplicity of infection (MOI) for serial dilutions was estimated by Poisson distribution; for example,~63% of cells are expected to be transduced at least once and thus survive selection with an MOI of 1.

To stably transduce TRIM5α variants, we chose an MOI of ~0.33 (25–30% survival during titering) to minimize multiple transductions per cell (<5% probability). CRFK cells were seeded in six-well plates at $2 \times 10^5$ cells/well the day prior to transduction; for deep mutational scanning libraries, sufficient cells were seeded to generate at least 500x independent transductions for each nucleotide variant (32 codons x 13 sites x 500 ÷ 25% survival = $8.3 \times 10^5$ cells). Cells were transduced at the appropriate MOI with 10 µg/mL polybrene and spinoculation (1100 x g, 30 min), then underwent 6 µg/mL puromycin selection starting at 24 hr post-transduction and continuing until untransduced controls were completely dead (usually ~7 days). Upon completion of selection, surviving cells were pooled and maintained in 2 µg/mL puromycin. Passages always maintained at least $5 \times 10^5$ cells (1000x library coverage) to avoid bottlenecking library diversity.

## Deep mutational scan, sequencing, and enrichment calculation

CRFK cells expressing a TRIM5α deep mutational scanning library were seeded in 12-well plates at $1 \times 10^5$ cells/well the day prior to viral infection. Sufficient wells were seeded for at least 1000x library coverage among target cells to be sorted 4 days later (assuming at least two doublings in that time, with sorting frequency typically ~5% of cells as estimated beforehand by viral titering against DMS library-expressing cells). Thus, each biological replicate began with at least $2.4 \times 10^6$ cells (32 codons x 13 sites x 1000 ÷ 5% ÷ 4) seeded from the same CRFK library.

Libraries were infected with HIV-1-GFP or N-MLV-GFP the following day. For loss-of-restriction experiments (*Figures 4–6*), we chose viral doses that were restricted by WT TRIM5α to <1%, as determined during preliminary titering experiments. For gain-of-HIV-1-restriction by human TRIM5α, we chose a viral dose in which WT TRIM5α was infected to ~98%, in order to minimize uninfected GFP-negative cells. Infection efficiency was monitored by parallel infection of controls (empty vector, WT TRIM5α, uninfected negative control). Cells were infected by spinoculation (1100 x g, 30 min) and media was replaced 24 hr post-infection. Infected cells were incubated an additional 48 hr to increase GFP expression levels. Cells were harvested by trypsinization, pelleted, and vigorously resuspended as well as filtered (0.7 µm) to minimize aggregation. Cells were FACS sorted, with stringent gating on size, single cells, and presence or absence of GFP (for loss- or gain-of-restriction, respectively). At least $4 \times 10^5$ cells (1000x library coverage) were sorted for each biological replicate. For gain-of-HIV-1-restriction by human TRIM5α, sorted GFP-negative cells were pelleted and re-seeded at $1 \times 10^5$ cells/well for a second round of infection, at the same dose, the following day, in order to enrich true restrictors and deplete cells uninfected by chance. Infection, harvest, and FACS sorting were all performed identically, except that apparent HIV-1 restriction by pooled variants was improved in the second round of enrichment (~50% GFP-negative compared to ~10% in the first round of infection). Sorted cells were pelleted, resuspended in PBS, and genomic DNA was harvested using DNeasy Blood and Tissue kits (Qiagen, Hilden, Germany). Input samples were harvested from infected but unsorted cells for each replicate.

Illumina libraries were constructed from genomic DNA by 2-step PCR amplification using Q5 polymerase. The first PCR amplified the v1 loop of TRIM5α and added adapters; the second set of PCR primers annealed to these adapters and added a unique 8 bp i7 Nextera barcode as well as P5 and P7 adapters for flow cell binding (see *Table 1*). Genomic DNA from each sample (two input replicates, two sorted replicates for each experiment) was amplified in three separate PCR tubes, with 500 ng of genomic DNA per tube, to offset random PCR jackpotting. This sampled a total of 1.5 µg of DNA, which represents ~500 x library coverage, assuming 6.6 pg gDNA/cell and a single TRIM5α integration/cell. After 15 cycles of amplification, samples were digested for 15 min at 37°C with 5 µL of ExoI (NEB) to remove first round primers. PCR products were then pooled from triplicate tubes, purified by QIAquick PCR purification kit (Qiagen), and the entire elution was divided between three separate PCR tubes for 18 cycles of second round amplification. Barcoded PCR products (234 bp) were pooled from triplicate tubes and purified by double-sided size selection using Ampure beads (Beckman Coulter, Pasadena, CA). In brief, large DNA was removed by incubation with 0.8x bead volume and magnetization; PCR products were bound from the supernatant with 1.5x bead volume, washed with 80% ethanol, and eluted in water. PCR product purity was confirmed by gel electrophoresis. Samples were then pooled at equimolar ratios and Illumina sequenced (MiSeq-v2) with single-end reads. Estimated read depth (equally distributed between libraries) always exceeded 1 million counts per library to ensure >1000 x coverage, on average, per nucleotide sequence. One read was generated using the i7 index primer for the 8 bp barcode, and a second read used a custom

sequencing primer, which annealed immediately adjacent to the v1 loop (33 bp read for human, 39 bp for rhesus TRIM5α). PhiX was included at 15% in sequencing runs to increase per-bp-diversity, since the majority (10/11 for human or 12/13 for rhesus) of reads should not randomize any given codon.

Reads counts for each unique nucleotide sequence from all four samples in an experiment were compiled into a single tsv file. Sequences that differed from WT by more than one codon, or sequences in which codons did not end in C or G, were filtered from the dataset; these largely had only a few reads per sample and represented sequencing errors. Reads counts were normalized to total counts per million (cpm) within each barcoded sample. Sequences with low read counts (<50 cpm) were excluded as they were found to introduce noise (poor correlation between replicates and across codons). Enrichment was calculated as the ratio of sorted to input cpm. The average and standard deviation of enrichment was calculated across both replicates of all synonymous codons to determine statistics at the amino acid level, except for WT variants, where we show each synonymous variant separately (averaged across replicates) to better visualize WT variance. Amino acid enrichment values were plotted in waterfall plots (descending order of enrichment), scatter plots (comparing replicates), and double-gradient heat maps (comparing amino acid variants at each position, with baseline value [white] set to the average for WT enrichment) using GraphPad Prism. R scripts for data analysis, including all filtering, normalization, and calculations, as well as raw sequence reads have been uploaded to Github: https://github.com/jtenthor/T5DMS_data_analysis (copy archived at https://github.com/jtenthor/T5DMS_data_analysis; *Tenthorey, 2020*).

## Calculation of fold virus inhibition

Viral inhibition by TRIM5α constructs was always compared to CRFK cells transduced with empty vector. CRFK lines were seeded in 96-well plates at $1 \times 10^4$ cells/well the day prior to transduction. Media was removed and replaced with serial 3-fold dilutions of the appropriate GFP-marked retrovirus. Serial dilutions were started at titers that yielded ~95% infection in untransduced CRFK. Plates were centrifuged at 1100 x g for 30 min and then incubated at 37°C. The following day, virus was removed and cells were fed fresh media. Cells were harvested by trypsinization 72 hr after transduction and analyzed by flow cytometry for GFP fluorescence. Cells were gated on size (FSC vs. SSC), single cells (FSC height vs. area), and GFP-positive as compared to negative control (FITC vs. PE empty channel).

Fold inhibition was calculated by comparing $ID_{10}$, the amount of virus required to infect 10% of cells, between TRIM5α and empty vector. Infection (% GFP-positive) was plotted against viral dose, both on logarithmic scale, as in *Figure 2E*. Infection points < 0.5% or > 50% GFP-positive were excluded due to increased noise or curve saturation, respectively, yielding a simple linear relationship between viral dose and % GFP-positive; infections without at least three data points in this range were excluded from further analysis. A linear regression (against log-transformed data) was then used to calculate the viral dose corresponding to 10% infection (back-calculated to linear scale), and the dose for TRIM5α was divided by that for empty vector. This method was used to calculate fold inhibition for all viruses except human TRIM5α against N-MLV, as we could not consistently achieve infection greater than 1% for WT human TRIM5α; we therefore report raw infection data. All fold inhibitions were calculated from at least three independent experiments, which were performed either in biological singlicate or duplicate.

## Immunoblot

CRFK cells stably expressing TRIM5α-HA variants were harvested by trypsinization, washed in PBS, and counted; $10^6$ cells were lysed for 15 min on ice in 100 μL pre-chilled lysis buffer (50 mM Tris, pH 8, 150 mM NaCl, 1% Triton-X100, 1x cOmplete EDTA-free protease inhibitor cocktail [Roche, Basel, Switzerland]). Lysates were pelleted at 20,000 x g for 15 min at 4°C. Supernatants were quantified by Bradford protein assay (BioRad) and normalized to load equal protein across all samples (usually 10–25 μg per lane). Samples were boiled for 5 min in Laemmli Sample Buffer (BioRad) supplemented with 5% β-mercaptoethanol and loaded onto Mini-PROTEAN TGX stain-free gels (BioRad). Gels were run in Tris/Glycine/SDS buffer (BioRad) for 50 min at 150 V, then transferred semi-dry for 7 min at 1.3 mV using Trans-Blot Turbo 0.2 μm nitrocellulose transfer packs and the Trans-Blot Turbo transfer system (BioRad). Blots were blocked with Odyssey blocking buffer (LI-COR, Lincoln, NE), then

probed with mouse anti-HA at 1:1000 (RRID:AB_2565335, Biolegend, San Diego, CA) and rabbit anti-β-actin at 1:5000 (RRID:AB_2305186, Abcam, Cambridge, UK). All antibodies were diluted in TBST with 5% bovine serum albumin (Sigma Aldrich, St. Louis, MO). Blots were washed in TBST and probed with IRDye 680RD donkey anti-mouse (RRID:AB_10953628, LI-COR) and IRDye 800CW donkey anti-rabbit (RRID:AB_621848, LI-COR), both diluted 1:10,000. Blots were washed and scanned at 680 and 800 nm. HA intensities were quantified using ImageJ and normalized to actin, then compared to WT TRIM5α to determine relative expression levels.

## TRIM5α phylogeny, rapid evolution analysis, and evolutionary accessibility

A tBLASTn search of human TRIM5α (NP_149023.2) against primate genomes returned 29 unique simian primate orthologs of TRIM5α. We excluded New World monkey sequences as they share a 9-amino acid deletion in the v1 loop. Open reading frames of the following sequences were translation aligned using MUSCLE: human (*Homo sapiens*, NM_033034.2), chimpanzee (*Pan troglodytes*, NM_001012650.1), bonobo (*Pan paniscus*, XM_003819046.3), gorilla (*Gorilla gorilla*, NM_001279549.1), Sumatran orangutan (*Pongo abelii*, NM_001131070.1), Bornean orangutan (*Pongo pygmaeus*, AY923179.2), white-handed gibbon (*Hylobates lar*, AY923180.1), white-cheeked gibbon (*Nomascus leucogenys*, NM_001280113.1), crab-eating macaque (*Macaca fascicularis*, NM_001283295.1), rhesus macaque (*Macaca mulatta*, NM_001032910.1), olive baboon (*Papio anubis*, NM_001112632.1), collared mangabey (*Cercocebus torquatus*, KP743974.1), sooty mangabey (*Cercocebus atys*, NM_001305964.1), drill (*Mandrillus leucophaeus*, XM_011971974.1), Wolf's guenon (*Cercopithecus wolfi*, KP743973.1), red guenon (*Erythrocebus patas*, AY740619.1), grivet (*Cercopithecus aethiops*, AY669399.1), tantalus monkey (*Chlorocebus tantalus*, AY740613.1), African green monkey (*Chlorocebus sabaeus*, XM_008019877.1), vervet monkey (*Chlorocebus pygerythrus*, AY740612.1), golden snub-nosed monkey (*Rhinopithecus roxellana*, XM_010364548.1), and Angola colobus (*Colobus angolensis*, XM_011963593.1).

A PHYML tree was built using the HKY85 substitution model with 100 bootstraps and rooted on human TRIM6 (NM_001003818.3). The unrooted tree was used for site-specific PAML analysis (*Yang, 1997*) using both F3 × 4 and F61 codon models to ensure robust results. We performed maximum likelihood (ML) tests comparing model 7 (neutral selection beta distribution) to model 8 (beta distribution with positive selection allowed). In each case, the model allowing positive selection gave the best fit to the data (p<0.0001, chi-squared test on 2x ΔML with two df). Model eight also identified rapidly evolving sites with a Bayes Empirical Bayes posterior probability >0.95. We report residues that meet this threshold for rapid evolution under both the F3 × 4 and F61 codon models.

Evolutionarily accessible amino acids were defined as one nucleotide substitution away from the wild-type sequence. For *Figure 1B*, we determined all amino acids that were accessible from any of the 22 aligned sequences, then determined the fraction of these amino acids that were represented in our alignment. Evolutionary landscapes for *Figure 1C–D* were generated from hypothetical data with 3D surface plots using the Plotly R package.

## Statistics

All statistics were performed using GraphPad Prism. Correlations between replicates or across experiments were computed using non-log-transformed data, and using non-parametric Spearman correlations where data did not pass normality tests. Student's unpaired t-tests were used to compare differences in mean values for normally distributed data from two groups; where variances were determined to be significantly different, Welch's correction was applied. For comparisons of more than two groups, one-way ANOVA tests were used with Holm-Sidak's multiple comparison correction; where variances were determined to be unequal, the Geisser-Greenhouse correction was applied. Exact p-values are reported in source data files.

## Acknowledgements

We thank all members of the Malik and Emerman labs, especially Shirleen Soh and Molly OhAinle, for feedback and advice. We thank Tera Levin, Kevin Forsberg, Molly Ohainle, Tyler Starr, Russell Vance, Patrick Mitchell, Janet Young, Nicholas Chesarino, Pravrutha Raman, Phoebe Hsieh, Shirleen Soh, and Rick McLaughlin for comments on the manuscript. The Fred Hutchinson Genomics Core

performed Illumina sequencing. Chimeric HIV-1 gag/pol vectors with SIV CA were a gift from Theodora Hatziioannou. HIV-1 and HIV-2 gag/pol vectors were a gift from Jeremy Luban. Work was supported by the Hanna H Gray fellowship (JLT), an EXROP award (AS), and an Investigator award (HSM) from HHMI, in addition to grants from the Mathers Foundation and NIAID (HARC [HIV Accessory and Regulatory Complexes] center, P50 AI082250, PI Nevan Krogan, subaward to ME, HSM). The authors declare no competing interests.

## Additional information

### Funding

| Funder | Grant reference number | Author |
|---|---|---|
| G Harold and Leila Y. Mathers Foundation | | Harmit S Malik |
| National Institute of Allergy and Infectious Diseases | P50 AI082250 | Michael Emerman<br>Harmit S Malik |
| Howard Hughes Medical Institute | Investigator award | Harmit S Malik |
| Howard Hughes Medical Institute | Hanna H. Gray fellowship | Jeannette L Tenthorey |
| Howard Hughes Medical Institute | EXROP award | Afeez Sodeinde |

The funders had no role in study design, data collection and interpretation, or the decision to submit the work for publication.

### Author contributions

Jeannette L Tenthorey, Conceptualization, Data curation, Software, Formal analysis, Supervision, Validation, Investigation, Visualization, Methodology, Writing - original draft, Writing - review and editing; Candice Young, Validation, Investigation, Visualization, Writing - review and editing; Afeez Sodeinde, Investigation; Michael Emerman, Harmit S Malik, Conceptualization, Supervision, Funding acquisition, Writing - review and editing

### Author ORCIDs

Jeannette L Tenthorey (iD) https://orcid.org/0000-0003-0175-080X
Michael Emerman (iD) https://orcid.org/0000-0002-4181-6335
Harmit S Malik (iD) https://orcid.org/0000-0001-6005-0016

### Decision letter and Author response

Decision letter https://doi.org/10.7554/eLife.59988.sa1
Author response https://doi.org/10.7554/eLife.59988.sa2

## Additional files

### Supplementary files

• Transparent reporting form

### Data availability

Sequencing data are available at the following GitHub repository: https://github.com/jtenthor/T5DMS_data_analysis (copy archived at https://github.com/elifesciences-publications/T5DMS_data_analysis).

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
