## [Decision Letter]

**Acceptance summary:**

In virus-host arms races, host antiviral proteins frequently evolve to counteract adaptive innovations in the virus. However, the time scales for host and viral evolution are quite discrepant, raising the question of how a slower evolving host can adapt its restriction factors to rapidly evolving viruses. Here, Tenthorey et al. use the antiviral restriction factor TRIM5α as a model to demonstrate that key regions of TRIM5 are resilient to high rates of mutagenesis, thus providing an evolutionary explanation for how hosts are able to "keep pace" with the viruses that infect them.

**Decision letter after peer review:**

Thank you for submitting your article "Mutational resilience of antiviral restriction favors primate TRIM5α in host-virus evolutionary arms races" for consideration by *eLife*. Your article has been reviewed by three peer reviewers, and the evaluation has been overseen by a Reviewing Editor and Detlef Weigel as the Senior Editor. The following individual involved in review of your submission has agreed to reveal their identity: Lionel Berthoux (Reviewer #3).

The reviewers have discussed the reviews with one another and the Reviewing Editor has drafted this decision to help you prepare a revised submission.

Summary:

Previous studies have used random mutagenesis to isolate mutants in the v1 loop of TRIM5α with increased antiviral function. Here, using infection-based functional screening of v1 loop mutant libraries, coupled with deep sequencing, the authors are able to generate a thorough analysis of mutations resulting in gain or loss of antiviral function. Key insights from the study are that (1) human TRIM5α exists in a "poised" sequence space, where missense mutations often exhibit potential for gain-of-function against diverse lentiviruses, and (2) that this sequence space is resilient to perturbation from missense mutations. This study contributes new insight to our understanding of the accessible sequence space of a rapidly-evolving region of a model restriction factor in the context of viral infection.

It is requested that the authors provide either new experimental data or additional scholarly discussion to address the following major criticisms of the study:

1) The premise of the article is that rapidly adapting residues in the v1 loop of TRIM5 α must have appeared repeatedly due to intrinsic mutational constraint. Upon testing this possibility, the authors found the opposite; TRIM5α is mutationally resilient in its v1 loop. The shortcoming of this submission is that there is a disconnect between the results generated by the artificial mutagenesis of TRIM5α and the actual evolutionary trajectory of TRIM5α in primates. Since loops in proteins are known to be generally flexible, it is not very surprising that the v1 loop of TRIM5α is mutationally resilient.

2) It would be helpful if the authors could provide clarity on the possible relationship between expression, charge, and anti-HIV1 activity. While the authors demonstrate that charge correlates with antiviral potency of human TRIM5α variants towards HIV-1 (Figure 3E) and that several LOF Rhesus TRIM5α variants express at WT levels (Figure 5C), it is still possible that in most cases charge of human TRIM5α influences expression/stability, which in turn influences antiviral potency. In Figure 3—figure supplement 1, the authors show that "some but not all" GOF mutations increase TRIM5α expression level – what is the Spearman for Figure 3—figure supplement 1B? Expression and potency still seem to correlate well, albeit with outliers.

3) Based on a small experiment combining R332P with a few other possible mutations, the authors conclude that the antiviral effects of mutations in the v1 loop are not additive. This is premature, as many other combinations of mutants could be tested. The lack of additive effects might be specific to the R332P mutant. Previous literature is also misquoted in this paragraph. Pham et al. (2010, 2013) did, in fact, report that some beneficial mutations were partly additive (R332G + R335G in particular).

4) As mentioned above, the findings are in stark contrast with the patterns of natural evolution in TRIM5α, where the v1 loop toggles between a select few amino acid residues at specific sites. Please provide more discussion of why the gain-of-function residues identified artificially do not occur naturally.

Revisions expected in follow-up work:

The editors are sensitive to the fact that labs may not be able to perform certain experiments in the current COVID-19 climate. The following comment would appear to require additional experiments. However, if the authors are able to address this point in the current manuscript by leveraging previous studies on known TRIM5 mutants that alter charge, that would be a reasonable alternative.

Specifically, there is no experimental consideration for lentiviral capsid sequence or capsid net charge to accompany the finding that the net charge of the v1 loop of TRIM5α is important for anti-lentiviral restriction. The authors claim that the mutational resilience of TRIM5α makes it more poised for success in host-virus arms races but they do not yet provide any experimental support for this statement. Please consider experiments that discuss this issue.

---

## [Author Response]

Revisions for this paper:It is requested that the authors provide either new experimental data or additional scholarly discussion to address the following major criticisms of the study:1) The premise of the article is that rapidly adapting residues in the v1 loop of TRIM5α must have appeared repeatedly due to intrinsic mutational constraint. Upon testing this possibility, the authors found the opposite; TRIM5α is mutationally resilient in its v1 loop. The shortcoming of this submission is that there is a disconnect between the results generated by the artificial mutagenesis of TRIM5α and the actual evolutionary trajectory of TRIM5α in primates. Since loops in proteins are known to be generally flexible, it is not very surprising that the v1 loop of TRIM5α is mutationally resilient.

We agree with the reviewers’ first point. Indeed, the disconnect between the mutationally permissive and resilient landscape we identify by DMS and the actual evolutionary history of TRIM5α is one of the most surprising and interesting results in our manuscript. As suggested in point 4 below, we have added additional discussion to the Discussion section to further describe hypotheses for how this difference might arise, such as additional selective constraints on TRIM5α that are independent of restriction of modern retroviruses and potential fitness costs to the host.

We wish to expand on the second point made by the reviewers, who correctly point out that protein loops have been shown to exhibit more mutational flexibility. However, this flexibility largely reflects the *structural* pliability of such loops, which can mutate without affecting the structure of the protein core. Most previously studied proteins did not rely on loops for critical functions like ligand binding. In contrast, TRIM5α is unusual in that its v1 loop is itself essential for ligand (capsid) binding. Therefore, we expected less mutational flexibility than previously studied loops due to *functional* constraints on a ligand-binding domain. Nevertheless, we found that the v1 loop is exceptionally resilient to mutation, which makes our finding very surprising and quite unusual. We have modified the Discussion to more clearly make this important point.

2) It would be helpful if the authors could provide clarity on the possible relationship between expression, charge, and anti-HIV1 activity. While the authors demonstrate that charge correlates with antiviral potency of human TRIM5α variants towards HIV-1 (Figure 3E) and that several LOF Rhesus TRIM5α variants express at WT levels (Figure 5C), it is still possible that in most cases charge of human TRIM5α influences expression/stability, which in turn influences antiviral potency. In Figure 3—figure supplement 1, the authors show that "some but not all" GOF mutations increase TRIM5α expression level – what is the Spearman for Figure 3—figure supplement 1B? Expression and potency still seem to correlate well, albeit with outliers.

To address this concern, we have added two panels to Figure 3—figure supplement 1C and D to assess whether or not decreased charge is linked to increased expression level. We find that decreasing the net charge of the v1 loop, at least among the variants we tested, always increases HIV-1 restriction but does not always increase expression. We clarify this point with the following text: “Indeed, reducing the electrostatic charge always improved HIV-1 restriction, but did not always increase TRIM5α expression level (Figure 3—figure supplement 1C-D).” We have also added text to the Discussion to describe how the detrimental effect of positive charge might largely be accounted for by lowering TRIM5α expression level (Figure 3—figure supplement 1C-D) and have cited additional references. Finally, as suggested by the reviewers, we have now also added a Spearman correlation (r = 0.80) to Figure 3—figure supplement 1B, which confirms our analysis that some, but not all, of the gain-of-function variants can be explained by increased expression levels.

3) Based on a small experiment combining R332P with a few other possible mutations, the authors conclude that the antiviral effects of mutations in the v1 loop are not additive. This is premature, as many other combinations of mutants could be tested. The lack of additive effects might be specific to the R332P mutant. Previous literature is also misquoted in this paragraph. Pham et al., 2010, 2013, did, in fact, report that some beneficial mutations were partly additive (R332G + R335G in particular).

We thank the reviewers for this correction. We have acknowledged this partially additive combination in the text: “Previous reports also found that most beneficial mutations are either non-additive or interfering, identifying only one combination (R332G with R335G) that was partially additive (Li et al., 2006; Pham et al., 2010; 2013).”

We also appreciate the reviewers’ point that we showed a lack of additive effects for double mutants only in combination with R332P. In fact, we think this is one of the most interesting features of the R332P mutation – this single mutation alone places human TRIM5α at a fitness peak! However, the reviewers correctly point out that we have not shown that other single mutations are already at their fitness peaks. We have therefore softened our conclusion to acknowledge this point, as follows: “Given that R332P is one of the strongest gain-of-function variants we identified, it remains possible that other, more modest gain-of-function variants might additively improve HIV-1 restriction. […] Thus, remarkably, human TRIM5α appears to be located only one mutational step away from fitness peaks in its evolutionary landscape of potential adaptation against HIV-1.”

4) As mentioned above, the findings are in stark contrast with the patterns of natural evolution in TRIM5α, where the v1 loop toggles between a select few amino acid residues at specific sites. Please provide more discussion of why the gain-of-function residues identified artificially do not occur naturally.

As discussed above (point 1), we agree with the reviewers that our findings are quite surprising in light of the constrained natural evolution of TRIM5α. The contrast between these patterns suggests that there are additional selective constraints driving TRIM5α evolution aside from the viral restriction activities tested in our paper. To acknowledge and speculate on those additional constraints, we have added extensive additional text to the Discussion.

Revisions expected in follow-up work:The editors are sensitive to the fact that labs may not be able to perform certain experiments in the current COVID-19 climate. The following comment would appear to require additional experiments. However, if the authors are able to address this point in the current manuscript by leveraging previous studies on known TRIM5 mutants that alter charge, that would be a reasonable alternative.Specifically, there is no experimental consideration for lentiviral capsid sequence or capsid net charge to accompany the finding that the net charge of the v1 loop of TRIM5α is important for anti-lentiviral restriction. The authors claim that the mutational resilience of TRIM5α makes it more poised for success in host-virus arms races but they do not yet provide any experimental support for this statement. Please consider experiments that discuss this issue.

We agree with reviewers that it would be interesting to examine the role of capsid charge on TRIM5α-mediated restriction, to investigate whether there is indeed electrostatic repulsion between TRIM5α and the capsid. Specifically, we would propose to mutate positively charged residues within the HIV-1 capsid to neutrally charged amino acids in the regions thought to interact with TRIM5. Indeed, we have plans to undertake such a study that have been significantly delayed due to COVID-19-related restrictions.

Although we agree that an experimental test of the mutational resilience hypothesis would be desirable, this is a significant undertaking to address appropriately. Ideally, one would compare TRIM5α for success in host-virus arms races with another restriction factor that did not display mutational resilience, for example by comparing proportion of viral escape mutations during in vitro viral evolution. Such a system does not exist at present, as we have not yet identified (or tested for) a restriction factor that lacks mutational resilience.